# Adversarial Attacks in Weight-Space Classifiers[*]

**Tamir Shor**  *tamir.shor@campus.technion.ac.il*
*Department of Computer Science*
*Technion – Israel Institute of Technology*
*Haifa, Israel*

**Ethan Fetaya**  *ethan.fetaya@biu.ac.il*
*Faculty of Engineering*
*Bar-Ilan University*
*Ramat Gan, Israel*

**Chaim Baskin**  *chaimbaskin@bgu.ac.il*
*School of Electrical and Computer Engineering*
*Ben-Gurion University of the Negev*
*Be'er Sheva, Israel*

**Alex Bronstein**  *bron@cs.technion.ac.il*
*Technion – Israel Institute of Technology, Haifa, Israel*
*Institute of Science and Technology, Austria*

**Reviewed on OpenReview:** *https://openreview.net/forum?id=eOLybAlili*

## Abstract

Implicit Neural Representations (INRs) have been recently garnering increasing interest in various research fields, mainly due to their ability to represent large, complex data in a compact, continuous manner. Past work further showed that numerous popular downstream tasks can be performed directly in the INR parameter-space. Doing so can substantially reduce the computational resources required to process the represented data in their native domain. A major difficulty in using modern machine-learning approaches, is their high susceptibility to adversarial attacks, which have been shown to greatly limit the reliability and applicability of such methods in a wide range of settings. In this work, we perform an in-depth security analysis of the behavior of weight-space classifiers under adversarial attacks. Our study reveals that parameter-space models trained for classification exhibit increased robustness to standard white-box adversarial attacks compared to standard classifiers that operate in the original signal space. This is achieved without the need of any robust training. We source this robust behavior to the phenomenon of gradient-obfuscation promoted during the INR optimization process, and pinpoint the limitations of this robustness under alternative adversarial approaches. To support our claims, we develop a novel suite of adversarial attacks targeting parameter-space classifiers, and furthermore analyze practical considerations of such attacks.

## 1 Introduction

Implicit Neural Representations (INRs) represent an arbitrary signal as a neural network that predicts signal values based on spatial or temporal coordinates. These representations have recently been raising increased interest due to their ability to efficiently and compactly encode high-dimensional data while maintaining high input fidelity, and have further been shown to be capable of learning additional intrinsic knowledge latent within the original signal-domain representation (Sitzmann et al., 2020; Wang et al., 2021). While

---

[*]Code available at https://github.com/tamirshor7/Parameter-Space-Attack-Suite.

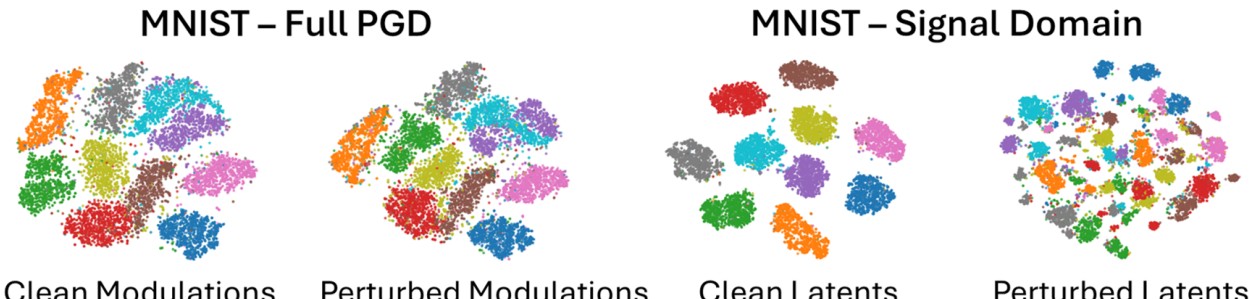

Figure 1: **t-SNE projection of modulation vectors fitted for clean and adversarially-perturbed data** - for parameter-space and signal-space classifiers.

INRs are inherently designed to fit one specific signal (rather than generalizing across an entire dataset), several recent works have proposed ways to use these representations to achieve a variety of downstream tasks by using some neural meta-network operating directly over the parameter-space of the INR of each signal within the dataset (Chen & Wang, 2022; Dupont et al., 2022; Lee et al., 2021; Navon et al., 2023b).

In these settings, each data sample is received in the native signal space, converted to its compressed INR representation by fitting a neural model, and finally the meta-network performs the designated downstream task over the implicit representation encoded by the model's parameters. These approaches alleviate challenges of processing high-dimensional data, often avoiding complex, specialized architectures and reducing computational demands.

Adversarial attacks are known to be a major concern for the use of neural networks across a variety of tasks. In this work, we explore – to the best of our knowledge, for the first time – the adversarial robustness of parameter-space classifiers. We start by framing the problem: any attack on a parameter-space classifier must be simultaneously optimized to reduce classification performance while also satisfy some constraint incurring the original, signal domain data fidelity. Since no prior studies have addressed this setting, we first formulate a novel set of possible white-box attacks over several data modalities. We then proceed to show that parameter-space classification models are inherently more robust to white-box adversarial attacks in comparison to the signal-domain classification problem counterpart.

We attribute the relative robustness observed in our experiments to a mechanism of gradient obfuscation inherent to the INR pipeline. Firstly, adversarial attacks commonly exploit the locally linear nature of modern deep classifiers in high-dimensional spaces to create perturbations that, while bounded, are heavily amplified by the classifier to create significant data manipulations in deeper layers (Ingle & Kulkarni, 2021; Romano et al., 2020). In the setting of parameter-space classification, however, an INR optimization loop is embedded into the inference process. This optimization is disjoint from the downstream objective and is dedicated to expressing global data features. We hypothesize that this process acts as a functional *scrubber*: due to the spectral bias of INRs towards low-frequency components (Sitzmann et al., 2020), the optimization reconstructs the global signal structure while effectively failing to fit the high-frequency adversarial perturbations. Consequently, the adversarial pattern is attenuated-or scrubbed-from the representation before it reaches the classifier.

Secondly, as we later show in section 4.5, performing adversarial attacks over parameter-space classifiers poses significant computational difficulties for a white-box attacker, as attacking the INR given a perturbed signal requires backpropagation through an internal optimization loop.

In summary, this paper makes the following contributions:

1. We propose several novel types of adversarial white-box attacks for parameter-space classifiers, balancing accuracy and computational resource consumption.

2. We analyze the behavior of parameter-space classifiers under adversarial attacks, and supply evidence of relative robustness of such classifiers to gradient-based white-box adversarial attacks (compared to signal-domain counterparts). We empirically study the origin of this inherent robustness, pinpointing it to gradient obfuscation originated from the internal INR optimization loop.

3. We explore and mark the limitations of this robust behavior under gradient-free attacks.

4. To substantiate our claims on 3D data, we also formulate a novel adversarial attack for voxel-grid data representations, and demonstrate its performance in both parameter-space and signal-space classification settings.

## 2 Related Work

The analysis of trained neural network parameters has raised increasing interest in recent years, evolving from statistical analysis to sophisticated deep learning on neural functionals. (Eilertsen et al., 2020) was one of the first works considering this idea, performing trained network parameter analysis to identify optimization hyperparameters. (Unterthiner et al., 2020) showed that trained CNN classifier parameters can be used to directly attain a good assessment of classification accuracy without running the full model's forward pass. Recent advancements have introduced equivariant architectures to process these parameters more effectively: (Navon et al., 2023a) proposed DWSNets to respect permutation symmetries in weight spaces, while (Zhou et al., 2023) and (Zhou et al., 2024) introduced Neural Functional Transformers (NFTs) and Universal Neural Functionals (UNFs) to leverage attention mechanisms for processing weights of diverse architectures.

(Dupont et al., 2022) were the first to demonstrate that neural network parameters, specifically of SIREN (Sitzmann et al., 2020) models, could be used as the direct and only input for training meta-learner models for classification, generative, and data-imputation tasks. This domain has expanded to generative modeling of weight spaces, where (Peebles et al., 2022) and (Schürholt et al., 2024) employ diffusion and hyper-representations to generate or manipulate model parameters.

In the context of adversarial robustness, our work relates to the phenomenon of gradient obfuscation and purification. (Athalye et al., 2018) famously categorized defenses relying on shattered or vanishing gradients—phenomena often observed in unrolled optimization loops—as causing a false sense of security. Similarly, (Yang et al., 2022) analyzed Deep Equilibrium Models (DEQs), finding that their fixed-point nature can mask gradients, necessitating implicit differentiation for effective evaluation. Conceptually, the "scrubbing" effect we observe parallels adversarial purification methods, such as DiffPure (Nie et al., 2022), which use generative processes to cleanse inputs. While (Luo et al., 2024) recently proposed training INRs specifically to enhance data robustness, our work uncovers the *inherent* robustness of parameter-space classifiers to white-box gradient-based adversarial attacks, that arises without such explicit robust training. Finally, regarding 3D data, adversarial attacks have primarily focused on point clouds (Xiang et al., 2019; Liu et al., 2019a) or neural radiance fields (Fu et al., 2023), leaving voxel-based parameter-space classification largely unexplored.

## 3 Method

### 3.1 Implicit neural representations

Let $\mathcal{X}$ denote a class of signals of the form $x : \mathbb{R}^m \to \mathbb{R}^n$ (for clarity, we consider the domain and the co-domain to be finite Euclidean spaces; a more general formulation is straightforward). Given a specific signal $x$, the fitting of an implicit neural representation consists of fitting a neural model $F_\theta$ parameterized by $\theta \in \mathcal{Y}$ so as to approximate $x$ in the sense of some norm, (Chen & Wang, 2022; Dupont et al., 2022)

$$\theta^*(x) = \underset{\theta \in \mathcal{Y}}{\arg\min} \quad \|F_\theta - x\|_{\mathcal{X}}. \tag{1}$$

In what follows, we denote the representation obtained by solving the minimization problem in equation (1) as a map $R : \mathcal{X} \to \mathcal{Y}$, so that $R : x \mapsto \theta^*(x)$. For the sake of clarity, we wish to stress that $F$ and $R$

are two related yet distinct functions - in our notation, $F$ is the INR projecting spatial conditioning onto signal values (e.g., if $F$ represents a signal x, $F$ maps pixel coordinates to pixel values at each coordinate). Conversely, $R$ is an optimization process of the INR given a signal. Namely, given a signal $x$, $R$ finds the best set of parameters $\theta$ (namely, the best INR $F$) to represent $x$.

### 3.2 Parameter-Space Classifiers

In contrast to conventional classifiers operating directly on the signal space $\mathcal{X}$, parameter-space classifiers operate on the signal INRs, $R(\mathcal{X}) \subset \mathcal{Y}$. We denote a parameter-space classifier as $M_\psi$, parameterized by $\psi$; the model receives the representation parameters $\theta = R(x)$ as the input and outputs the predicted class label $\hat{c} = M_\psi(\theta)$ as the output.

Even for small neural representation models, the parameter space $\mathcal{Y}$ is often high-dimensional, increasing difficulties related to computational resources and overfitting when training the classification models on it (Navon et al., 2023a;b). Functas (Dupont et al., 2022) cope with this issue of increased parameter-space dimensionality by partitioning the parameter space into shared weights and sample-specific biases. Namely, instead of optimizing an entire distinct MLP model for every represented signal (e.g. image) from the dataset, Functas optimize a single MLP for the entire dataset. For every specific sample in the dataset, an INR is optimized by maintaing the global set of weights, and only optimizing the set of biases to present each distinct data sample. Mathematically, denoting the entire set of weights of each INR $\mathcal{Y}_\mathrm{w}$ and the entire set of biases $\mathcal{Y}_\mathrm{b}$ (so that $\mathcal{Y} = \mathcal{Y}_\mathrm{w} \cup \mathcal{Y}_\mathrm{b}$), all samples share a single vector of weights $\theta_\mathrm{w} \in \mathcal{Y}_\mathrm{w}$ across a collection of represented signals, while allowing to fit individual *modulation vectors* $v(x) \in \mathbb{R}^d$ of a fixed dimension with a learned shared linear decoder $W$ such that each signal has an individual vector of biases, $\theta_\mathrm{b}(x) = Wv(x)$. This technique was successfully demonstrated with using the SIREN (Sitzmann et al., 2020) models as the MLP backbone for neural representations.

As previously mentioned, $\theta$ relates to the entire set of parameters of the INR $F$. For SIRENs (used in this paper), $\theta$ is composed of a set of weight parameters $\theta_w$, shared across all data samples within a dataset, and a set of bias parameters $\theta_b$, that are distinct for each represented signal. Nonetheless, for convenience (and with slight abuse of notation) for the rest of the paper we denote parameters $\theta_b$ governing the representation of every $x$, as $\theta$.

### 3.3 Threat Model

In this work we explore a setting where the adversary operates in the signal domain $\mathcal{X}$, but the classification occurs in the parameter domain $\mathcal{Y}$. The rational for this choice, rather than applying perturbations in parameter-space, is two-fold. First, bounding attacks in parameter-space is difficult to define. The signal-domain response of an adversarial perturbation applied in weight-space is difficult to evaluate and define in such manner that applies to a wide-enough set of neural representations. We defer the study of this matter to future work. Second, the core of our work is the exploration of adversarial robustness of weight-space classifiers in comparison to signal-space classifiers. For a fair comparison, we perturb both types of pipelines in signal domain.

**Adversary Goal** - We consider an *untargeted evasion* setting. Given a clean input signal $x \in \mathcal{X}$ with a ground-truth label $y$ (or the model's clean prediction $\hat{y} = M_\psi(R(x))$), the adversary aims to find a perturbation $\delta$ such that the parameter-space classifier $M_\psi$ misclassifies the implicit representation of the perturbed signal. Formally, the adversary seeks to maximize a loss objective $\mathcal{L}$ (e.g., Cross-Entropy):

$$\max_{\delta} \mathcal{L}\left(M_\psi(R(x + \delta)), y\right) \tag{2}$$

subject to the perturbation constraints defined below.

**Adversary Capabilities** - The adversary is capable of modifying the data input, but is constrained by the nature of the INR pipeline.

- **Signal-Domain Constraint:** The adversary can only add perturbations $\delta$ to the input signal $x$ in its native domain. The perturbation is bounded by a norm budget $\|\delta\|_p \leq \epsilon$ (e.g., $L_\infty$ or $L_0$ for voxel grids) to ensure the attack remains imperceptible or physically realizable.

- **Indirect Parameter Manipulation:** Crucially, the adversary **cannot directly manipulate** the INR parameters $\theta$. The adversarial parameters $\theta_{adv}$ are a derived quantity resulting from the modulation optimization process $R$. Thus, the adversary must solve a bi-level optimization problem: finding a $\delta$ in signal space that, after passing through the non-linear optimization mapping $R(x + \delta)$, results in a $\theta_{adv}$ that lies in the error region of $M_\psi$.

**Adversary Knowledge** - We assume a **White-Box** setting, where the adversary has complete transparency regarding the system components. Specifically, the adversary has full knowledge of the classifier weights $\psi$, the INR architecture, and the specific algorithm used for $R$ (including the optimizer, learning rate, and number of steps $n$). While the adversary theoretically possesses the information required to compute exact gradients via backpropagation through time (BPTT), the optimization process $R$ may involve hundreds of steps, making full gradient computation computationally expensive or memory-prohibitive. Consequently, our evaluation considers adversaries with varying computational budgets (including the full budget), utilizing both approximate methods (e.g., section 3.4.2) and exact methods (e.g. sections 3.4.1, 3.4.5).

**Defense Setting** - The defender employs a standard parameter-space classifier $M_\psi$ trained on clean modulation vectors. We explicitly **exclude** adversarial training or preprocessing defenses from the defender's strategy. This ensures that our results measure the inherent robustness properties of the parameter-space architecture itself, rather than the efficacy of a specific defense algorithm.

## 3.4 Parameter-Space Adversarial Attacks

Given a pre-trained parameter-space classification model $M_\psi$, our objective is to optimize an adversarial perturbation $\delta$ in signal domain under some $L_q(\mathcal{X})$-norm bound,

$$\max_\delta \quad d(M_\psi(R(x + \delta)), M_\psi(R(x))) \text{ s.t. } \|\delta\|_{L_q(\mathcal{X})} \leq \epsilon, \tag{3}$$

where $d$ denotes a distance in the class label domain.

Note that while the signal is attacked in its native domain, the classifier $M$ accepts its input in the representation domain. The two domains are related by the map $R$ – a function whose pointwise value requires the solution of an optimization problem via a numerical algorithm. While backpropagating through this optimization process is possible (e.g. via automatic differentiation or implicit differentiation (Navon et al., 2020)), as we later show in Section 4, doing so dramatically increases computational resources required for perturbation optimization, thus posing additional practical challenges for a potential attacker, in addition to the inherent difficulty of optimizing such an attack that stems from the inference pipeline itself, as motivated in section 1.

In the following sections, we propose five different adversarial attacks, addressing these challenges that are absent in the traditional signal-domain adversarial attack settings.

### 3.4.1 Full Projected Gradient Descent (PGD)

A straightforward approach to solving equation (3) is by performing projected gradient descent (PGD) and backpropagating through both the classification model $M$ and the modulation optimization steps collectively denoted $R(x)$. While backpropagation through $M_\psi$ can be performed based on common first-order differentiation methods, a potential attacker must allow gradient flow through the modulation optimization steps performed in the forward pass of $R$. This requires differentiation through those optimization steps, namely via second-order differentiation and optimization. As we further show in section 4.5, this may substantially increase the amount of computational resource required for the attacker.

The full PGD attack pipeline is portrayed in figure 2 - in each PGD iteration, the current perturbation is added to the original signal. Then, the perturbation is kept frozen and the perturbed signal is being optimized for the corresponding INR $\theta^*$, that is being fed into the parameter-space classifier $M$.

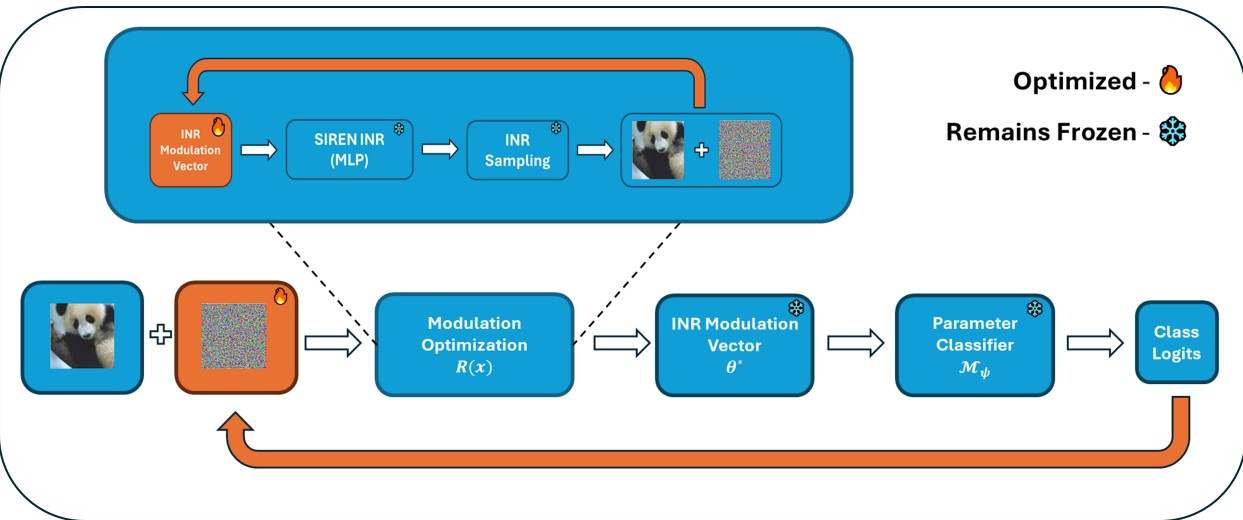

Figure 2: **Parameter-space classifier adversarial attack pipeline** - for a single PGD iteration. Orange blocks are optimized, blue blocks remain frozen.

### 3.4.2 Truncated Modulation Optimization (TMO)

Given the potentially immense computational resources required for full PGD attack optimization, we explore several attacks bypassing the need for full second-order differentiation in perturbation optimization – In *Truncated Modulation Optimization* (TMO) we apply a principle similar to the Truncated Backpropagation Through Time algorithm, widely used in RNN optimization. Since computational resources required for full PGD linearly grow with the number of modulation optimization steps $n$, in this attack we fix a certain parameter $\tau$ controlling the number of modulation optimization steps through which gradients would flow in attack optimization. Namely, rather than performing the forward pass of $R$ consisting of $n$ modulation optimization iterations, only $\tau$ optimization steps are performed, all of which will be backpropagated through. Since the full number of modulation optimization steps $n$ is inherent within the attacked model, an adversarial perturbation optimized for a duration of $\tau$ modulation steps might fail to fool the classifier at inference time (even if the attack succeeded after $\tau$ steps). To alleviate this gap, when optimizing an adversarial perturbation $\delta$ for a given data sample $x$, we first run the full $n$ optimization steps of $R(x)$ to find the "clean" modulation vector. In every PGD iteration to follow, the $\tau$ optimization steps are initialized from the previously optimized "clean" modulation.

### 3.4.3 Backpropagation Over Truncation Through Optimization of Modulation (BOTTOM)

TMO (section 3.4.2), while computationally light, suffers from the limitation of performing fewer modulation optimization steps per every PGD iteration. Model behavior observed by the attacking algorithm therefore deviates from the full forward process of the attacked model. In *Backpropagation Over Truncation Through Optimization of Modulation* (BOTTOM) we address this, and propose an attack that performs the full $n$ modulation optimization steps per every sample $x$, while also alleviating the computational prices described in section 3.4.1. For every PGD iteration, the $n$ steps performed in the forward pass of $R$ are divided into $\lfloor \frac{n}{\tau} \rfloor$ optimization segments. In each segment, second order differentiation is performed through $\tau$ modulation optimization steps.
The attacker is given the flexibility to balance between computational resource consumption (lower $\tau$ values) and fidelity of gradients to the full forward pass in $R$ (higher $\tau$ values).

### 3.4.4 Imposition of Constraints via Orthogonal Projection (ICOP)

Proposed attacks in previous sections focus on perturbations applied in signal domain. In this section we further propose an attack applied in INR domain. The main challenge here is efficiently constraining the

adversarial perturbation - while we perturb the neural representation of the signal, we still must constrain the response of every applied perturbation in signal domain. Namely, unlike the formulation in equation 3, here we must impose $F(\delta) \leq B$. The need to query the Functa model $F$ to assess this response makes it difficult to apply any trivial, immediate projection of a given perturbation $\delta$ onto the feasible set. We therefore propose *Imposition of Constraints via Orthogonal Projection* (ICOP) - this method relies on two approaches to impose soft constraints over the perturbations - First, after every PGD step we compute a projection loss, measuring constraint violation in $\mathcal{L}_1$: $\mathcal{L}_{proj.} = \|\mathcal{L}_q(F(\delta)) - B\|_1$. We then iteratively take small gradient steps $\eta$: $\delta_{proj.} = \delta - \eta \frac{\partial \mathcal{L}_{proj.}}{\partial \delta}$, until constraint satisfaction.

Secondly, to discourage perturbation steps from severely breaking signal-domain constraints, we also found it useful to project the maximized classification loss to be orthogonal to $\frac{\partial \|F(\theta+\delta)-F(\theta)\|_1}{\partial \delta}$ (namely, the perturbation-induced deviation in signal-domain) in every perturbation gradient step. We enforce soft-constraints in optimization, and then impose hard constraints over the optimized perturbation to ensure feasibility. This is done by computing $R$ over $F(\delta)$ clamped onto the feasible set.

### 3.4.5 Implicit Differentiation

Implicit differentiation provides a principled mechanism for computing gradients in bilevel optimization problems, where the output of an inner optimization procedure is implicitly defined by the stationarity conditions of an objective function. In our adversarial setting (equation 3), although the perturbation $\delta$ is applied directly in the signal domain, the classification loss depends on the output of the implicit representation mapping $R$, whose evaluation itself requires solving an optimization problem (equation 1). Explicit differentiation through this process (e.g., via unrolling or backpropagation through time) leads to computational graphs whose size scales linearly with the number of modulation optimization steps, incurring substantial memory and runtime costs (sections 3.4.1–3.4.3).

To mitigate this overhead, we additionally investigate an implicit differentiation (ID) baseline. Let $d(x, \delta)$ denote the adversarial classification loss (as defined in equation 3), and let $\mathscr{L}(\theta, x)$ denote the INR reconstruction objective minimized in equation 1. Assuming that the inner optimization is solved to a local optimum $\theta^* = R(x)$, the optimal parameters satisfy the first-order stationarity condition

$$\nabla_\theta \mathscr{L}(\theta^*, x) = 0.$$

Implicit differentiation exploits the fact that this condition must remain satisfied under an infinitesimal perturbation of the input. Specifically, for an infinitesimal change $dx$ in the signal and the corresponding change $d\theta$ in the optimal parameters, the perturbed solution must satisfy

$$\nabla_\theta \mathscr{L}(\theta^* + d\theta, x + dx) = 0. \tag{4}$$

A first-order Taylor expansion of this expression around $(\theta^*, x)$ yields the linear system

$$\underbrace{\nabla^2_{\theta\theta} \mathscr{L}(\theta^*, x)}_{\text{Hessian}} d\theta \ + \ \nabla^2_{\theta x} \mathscr{L}(\theta^*, x) \, dx \ = \ 0.$$

Solving for $d\theta$ and applying the chain rule gives the implicit gradient of the adversarial objective with respect to the perturbation:

$$\frac{\partial d}{\partial \delta} = -\frac{\partial d}{\partial \theta^*} \cdot (\nabla^2_{\theta\theta} \mathscr{L})^{-1} \cdot \nabla^2_{\theta x} \mathscr{L} \cdot \frac{\partial(x+\delta)}{\partial \delta}. \tag{5}$$

Equation 5 avoids explicit evaluation of $\frac{\partial \theta^*}{\partial x}$ and therefore eliminates the need to backpropagate through the modulation optimization trajectory used to solve equation 1. As a result, the memory footprint of this attack remains constant with respect to the number of modulation optimization steps, in contrast to the explicit differentiation-based attacks presented in sections 3.4.1–3.4.4.

**Validity and limitations** - The implicit formulation relies on two standard assumptions: (i) *stationarity*, namely that the inner optimization is solved sufficiently close to a local optimum such that $\|\nabla_\theta \mathscr{L}\| \approx 0$, and

(ii) *invertibility* of the Hessian $\nabla^2_{\theta\theta}\mathscr{L}$ at $\theta^*$. In practice, INR optimization is performed for a finite number of steps and typically does not reach exact stationarity, while the Hessian may be ill-conditioned or singular due to the non-convexity of the objective. In our implementation, we address the latter by applying damped identity regularization when computing inverse-Hessian vector products.

Importantly, implicit differentiation yields gradients corresponding to the *fixed-point solution* of the inner optimization, rather than to the finite, truncated optimization trajectory actually executed during inference. As we demonstrate empirically in Section 4.1, this mismatch limits the effectiveness of implicit gradients for adversarial optimization in our setting, causing this baseline to underperform explicit unrolling-based attacks despite its favorable computational properties.

### 3.5 BVA - Voxel-Grid Space Adversarial Attack

To study adversarial attacks over 3D structures, we must choose how to represent them, so that we can establish appropriate attack perturbation and constraints. Out of the 3 most common conventions for 3D data representations – voxel-grids, mesh-grids and point-clouds – in this work we specifically study adversarial perturbations over voxel-grid data. The reason for this choice is that our experiments show that due to the highly irregular structure of mesh-grids and point-clouds, fitting the dataset-shared set of INR weights ($\theta_w$ in notation from section 3.1) yields poor results, preventing achieving implicit representations of adequate signal fidelity.

While adversarial attacks over 3D have been studied in the past for mesh-grids (Xu et al., 2022) and point clouds (Liu et al., 2019a), to the best of our knowledge no similar work had so far been done in the context of voxel-grid representations. Since we wish to perform adversarial attacks over INR classifiers of voxel grids, and compare robustness to a corresponding signal-domain classifier, we develop Binary Voxel Attack (BVA) - a novel adversarial attack applied over voxel-grid data.

Voxel-grid representations are traditionally binary spatial occupancy maps. Therefore, an adversarial attack performing addition of a real-valued adversarial perturbation bounded in $\mathcal{L}_\infty$ is not applicable. Since our attacked signal is a binary map, we opt to establish an adversarial attack in the from of bit flipping (namely, bit attack). Given a voxel-grid input signal $x \in \{0,1\}^{H \times W \times D}$ (where $H, W, D$ are grid spatial dimensions), we formulate our attack as:

$$\max_\delta \quad \|M_\psi(x \oplus \delta)\| \text{s.t.} \|p\|_0 = B \tag{6}$$

where $\delta \in \{0,1\}^{H \times W \times D}$ is the optimized binary adversarial perturbation, and $\oplus$ denotes the bit-wise xor operation (performing bit switch over $x$ entries where $\delta$ has logical value of 1). $\delta$ is chosen to be bounded in $\mathcal{L}_0$ norm, namely bounding the number of bits flipped in $x$.

This formulation calls for the incorporation of binary optimization techniques, as trivial binarization of $\delta$ (e.g. by thresholding) is non-differentiable. To optimize $\delta$ as a binary mask we follow the binary optimization technique presented in (Iliadis et al., 2020). This method maintains two congruent masks - a binary mask $\Phi$ and a real mask $\Phi_c$. $\Phi$ is used for forward and backpropagation, where each gradient step applies the gradients calculated for $\Phi$ over $\Phi_c$. The latter is then binarized (without gradient tracking) to produce $\Phi$ for the next iteration.

To demonstrate relative robustness of parameter-space classifiers over voxel-grid data, in section 4.2.1 we first exemplify the efficacy of signal-domain attacks using BVA. Then, in section 4.2.2 we employ BVA in parameter-space attacks and demonstrate increased adversarial robustness to gradient-based attacks. We emphasize that equation 6 is a specific instance of equation 3, and therefore all parameter-space attacks previously proposed in section 3 can be performed on voxel-grids using BVA.

## 4 Results

In the following sections we demonstrate the relative robustness of Parameter-Space classifiers over 3 different datasets - MNIST (Deng, 2012), Fashion-MNIST (Xiao et al., 2017) and ModelNet10 (Wu et al., 2015). MNIST and Fashion-MNIST are chosen to represent the prevalent and heavily-studied modality of 2D images. While current adversarial robustness and computer-vision literature tends to focus on more challenging and

diverse datasets (e.g. CIFAR100, ImageNet), applicability of these datasets in the parameter-space classification setting is currently limited. This is because state-of-the-art parameter-space classification algorithms proposed thus far have yet to achieve classification satisfactory classification accuracies on such complex datasets ( (Navon et al., 2023a;b; Chen & Wang, 2022). ModelNet is used to demonstrate our findings over one of the modalities where usage of INRs is naturally useful, due to commonly large dimensionality. Following the approach from (Dupont et al., 2022), for each dataset $\mathcal{X}$ we first optimize a set of modulation vectors $\{v_x \forall x \in \mathcal{X}\}$. Then, we train a single MLP classifier over this modulation dataset. Finally, every pre-trained classifier is attacked with each of the five attacks presented in section 3.4. We emphasize no robust or adversarial training is being done in any stage of this experimentation scheme. Attack formulations and results for 2D data are presented in section 4.1, and for 3D data in section 4.2. In section 4.5 we further demonstrate the sense of adversarial robustness stemming from the added computational costs induced by performing adversarial attacks over deep weight-space classifiers, posing added practical challenges for potential attackers. Full training details are further specified in the appendix.

## 4.1 Robustness For 2D Data

We evaluate the adversarial robustness of parameter-space classifiers on MNIST and Fashion-MNIST across all attacks in our proposed suite. To ensure comparison against a strong and widely accepted baseline, that is not reliant solely on white-box PGD attacks, we also assess robustness under the Auto-Attack framework Croce & Hein (2020b). Clean and adversarial INR classification accuracy over 2D data classifiers is shown in table 1. In the notations of equation 3, $q = \infty$ is chosen as $B = \frac{\varepsilon}{256}$ for every table entry. *Clean* column represents classifier accuracy without introducing any adversarial perturbations.

Results indicate four major trends - First, increased adversarial robustness is measured for both MNIST and Fashion-MNIST, where only Auto-Attack used with the largest considered attack bound manages to severely harm classification accuracy, to an extent similar to that commonly observed in adversarial attacks over conventional classifiers operating in signal-domain. For MNIST, while the Auto-Attack, TMO and BOTTOM attacks are the only attacks that managed to incur any above-marginal decrease in classification accuracy, it is still markedly lower than the accuracy decrease of around 60% that is traditionally observed in similarly-bounded PGD attacks over signal-domain classifiers without any robust training (Wong et al., 2020). While higher-bounded attacks over the Fashion-MNIST dataset have been more successful, we claim their success is substantially lower than the effect of a similarly constrained attack over a signal-domain classifier.

To show this, we train four baseline signal-domain classifiers and attack with PGD using similar $\mathcal{L}_\infty$ norm constraints and PGD iteration number. We use Vision-Transformers (ViTs) (Dosovitskiy, 2020) due to their outstanding record in natural image classification (Han et al., 2022), and two convolutional models for their ubiquity and foundational role in computer-vision. To experiment with models of different sizes we train a ViT-Large model and a Resnet18, as well as a ViT-Base model and a compact, 2-layer CNN. Full architecture details are included in the appendix. We compare the robust accuracy of these four Signal-space classifiers under standard 100-iteration PGD attack, with that of our parameter-space classifier subjected to all attacks from section 3.4. Results, shown in figure 3, show parameter-space classifiers exhibit significantly higher robustness under white-box attacks, especially for lower attack bounds. These results highlight the relative robustness of parameter-space classifiers to white-box adversarial attacks. Under Auto-attack (that does not rely strictly on gradients and white-box attacks) this trends is maintained, however the absolute robustness of parameter-space models is lessened. As further shown in Section 4.4, this trend is likely caused by gradient-obfuscation appearing in gradient-based attack optimization over weight-space classifiers.

The second indicated trend is that the full PGD attack demonstrates lower adversarial potential compared to BOTTOM and TMO attacks in all experiments. We attribute this to the fact that in full-PGD the numerous steps of second-order optimization required for standard PGD in our case encourage vanishing gradients phenomena, namely attenuating gradient information backpropagated through the modulation optimization process $R$. These difficulties are added to the computational encumbrance of running full PGD (see section 4.5), and further stress the need in initial methods developed here (sections 3.4.2&3.4.3), as well as further related research. While computationally lightest, implicit differentiation proves less potent than most explicit methods. We attribute this to the strong dependence of this method in fulfillment

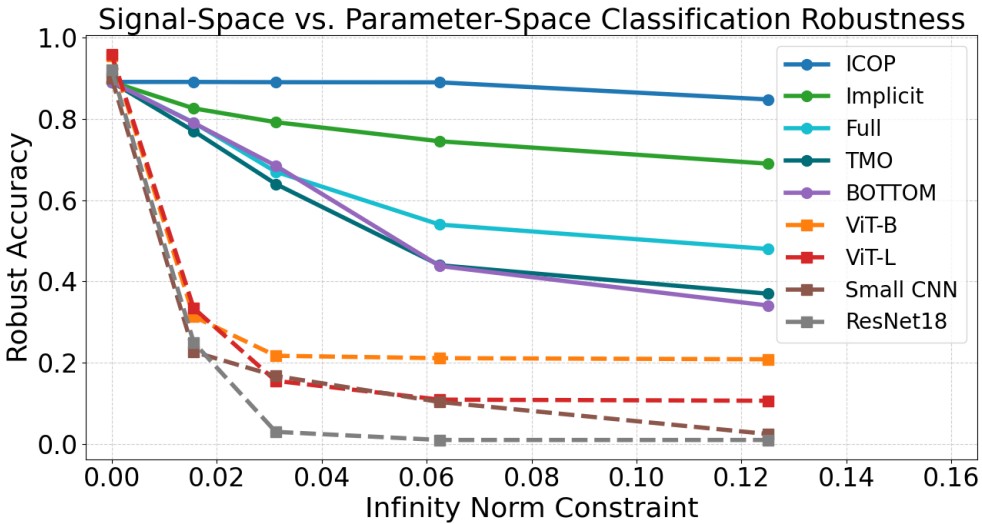

Figure 3: **Robust accuracy over the Fashion-MNIST dataset under $\mathcal{L}_\infty$ norm constraints** - for parameter-space (solid curves) and signal-space (dotted curves) classifier attacks.

of specified optimality conditions and strict stationarity assumptions required by the implicit function theorem (i.e., $\nabla_{\theta^*}\mathcal{Z} \approx 0$). In realistic adversarial settings constrained by finite optimization budgets, these KKT conditions are rarely fully satisfied. Consequently, explicit unrolling provides a more faithful gradient estimate of the actual optimization trajectory compared to the theoretical fixed-point approximation assumed by implicit differentiation.

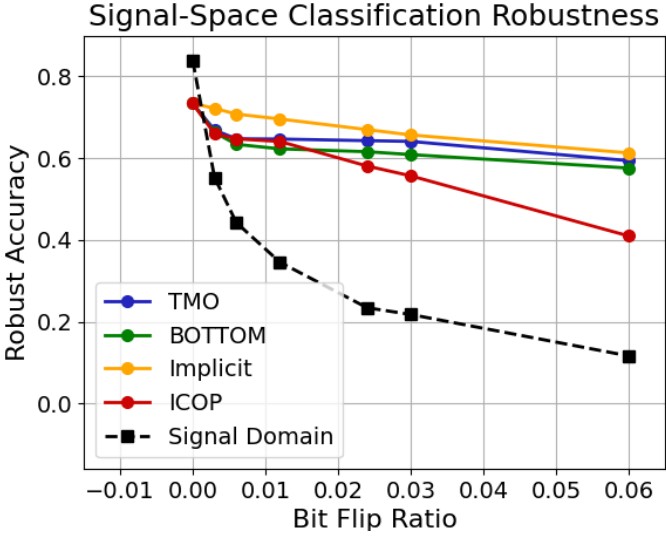

Figure 4: **BVA robust accuracy for signal-space classifier and weight-space classifier** - over the ModelNet10 dataset, under $\mathcal{L}_0$ norm constraints.

Another observation is that our proposed attack suite outperforms Auto-Attack in diminishing classification accuracy under almost all considered attack bounds. This advantage of our suite over Auto-Attack accompanies the substantially-lower ($40\times$ faster) optimization times required for optimization of our proposed attacks (see appendix for the complete analysis), that could potentially hinder the use of Auto-Attack in

our setting (even in cases where it is theoretically more-potent). Lastly, we conclude that the INR-domain attack (ICOP) failed to impose any actual effect on classification accuracy. We attribute this result to the heuristic nature of the approaches used to constrain the INR-domain perturbation's response in the signal domain (section 3.4.4). We still include it within our suite of proposed attacks, as it proved more efficient when initiated over 3D data (section 4.2).

Table 1: **MNIST & Fashion-MNIST Clean/Robust Classification Accuracy** - for any $\varepsilon$ value, $\mathcal{L}_\infty$ attack bound (equation 3) is $B = \frac{\varepsilon}{256}$.

| $\varepsilon$ | MNIST | | | | | | | Fashion-MNIST | | | | | | |
|---|---|---|---|---|---|---|---|---|---|---|---|---|---|---|
| | Clean | Full | TMO | BOTTOM | ICOP | Implicit | Auto-Attack | Clean | Full | TMO | BOTTOM | ICOP | Implicit | Auto-Attack |
| **4** | 0.976 | 0.949 | 0.941 | 0.939 | 0.965 | 0.957 | 0.968 | 0.891 | 0.791 | 0.77 | 0.79 | 0.891 | 0.826 | 0.808 |
| **8** | 0.976 | 0.915 | 0.91 | 0.906 | 0.965 | 0.946 | 0.948 | 0.891 | 0.67 | 0.64 | 0.658 | 0.8904 | 0.792 | 0.645 |
| **16** | 0.976 | 0.866 | 0.825 | 0.821 | 0.961 | 0.926 | 0.902 | 0.891 | 0.54 | 0.44 | 0.438 | 0.89 | 0.745 | 0.351 |
| **32** | 0.976 | 0.803 | 0.7035 | 0.698 | 0.957 | 0.919 | 0.44 | 0.891 | 0.48 | 0.37 | 0.341 | 0.848 | 0.69 | 0.05 |

## 4.2 Robustness For 3D Data

In section 4.1 we survey adversarial robustness of classifiers INR for 2D data. The importance of this baseline is comparability to the widely studied adversarial robustness of 2D classifiers. Nonetheless, practical advantages of using INRs for 2D data are usually limited, since common grid representations suffice for most common tasks. Data modalities of larger dimensions, such as 3D data, however, more commonly gain from using INRs for achieving downstream tasks, including classification (Jiang et al., 2020; Molaei et al., 2023; Niemeyer et al., 2020). We use the ModelNet 10 dataset (Wu et al., 2015) consisting of 4900 3D CAD models given in Mesh-grid representation.

### 4.2.1 Signal Space Attack

To demonstrate relative adversarial robustness over 3D data under the BVA attack (section 3.5), we must first demonstrate its prowess in attenuating signal-domain classification accuracy. To do so we train a voxel-grid classification model based on Point-Voxel CNN layers proposed by (Liu et al., 2019b). We train the classifier over ModelNet10, converting every mesh-grid sample to a voxel-grid of dimension $15 \times 15 \times 15$ using Trimesh (Dawson-Haggerty et al.). Then, we attack this trained model with BVA for a wide range of $\mathcal{L}_0$ bounds and assess attacked classification accuracy. Results are displayed in figure 4, showing BVA is efficient over the tested signal-space classifier, inducing exponential decay of classification accuracy with growth in number of flipped voxel bits.

### 4.2.2 Parameter Space Attack

Figure 4 compares the robust accuracy of the signal domain (black) and parameter-space classifiers, where the parameter-space classifier is attacked with TMO, BOTTOM, ICOP, and implicit attacks. We note that we do not experiment with the full PGD attack in parameter-space or with Auto-Attack for the 3D INR classifier, since to optimize INR for the ModelNet10 dataset with sufficient input signal fidelity, we performed 500 modulation optimization iterations per sample. While computation times for doing so are not very high (several seconds), backpropagation through these 500 optimization steps calls for infeasible memory and compute time requirements, and also lead to severe vanishing/exploding gradient issues. We further elaborate on these considerations in the appendix.
The data from figure 4 is presented numerically in the appendix.

## 4.3 Qualitative Results

In this section we provide qualitative evidence for the relative robustness of parameter-space classifiers. Adversarial attacks are known to effectively exploit the amplifications of small input-space perturbations (Goodfellow, 2014; Haleta et al., 2021). We attribute the robustness of parameter-space classifiers to the attacker's inability to directly perturb the classifier's input, instead requiring a constrained signal-domain perturbation that remains effective after modulation optimization (equation 1). This is deemed to be sub-

stantially more challenging for the attacker, as modulation optimization tends to attenuate adversarial effects fed as input to the classifier (figure 5).

To illustrate this phenomenon, we report the mean amplification, measured as the difference in activation values between perturbed and clean input data, for each layer in our pipeline (figure 2). These measurements, shown in figure 5, were calculated using the MNIST dataset, with every sample propagated through the pipeline after being perturbed by each attack in our suite with a constraint of $\varepsilon = \frac{32}{256}$. To further validate the relationship between attack success and amplification, we differentiate between robust and non-robust samples.

Successful attacks demonstrate an average amplification approximately three times greater than that of failed attacks. In both cases, amplification sharply plummets upon entering the first classifier layer (layer 10) and remains consistently low in subsequent layers. These findings indicate that adversarial amplification within the INR layers has a negligible impact on the downstream classification model. This supports our claim that modulation optimization effectively obfuscates the adversarial potency of perturbations applied in the signal domain. Importantly, this effect is not observed for signal-domain classifiers, as we show in figure 6. This figure illustrates the mean amplification difference measured through model activation layers over MNIST data, similarly to figure 5 for the signal-domain classifier trained according to section A.3.2. The adversarial amplification is not attenuated, but rather drastically inflated. Results for other attack constraints and datasets are reported in the appendix.

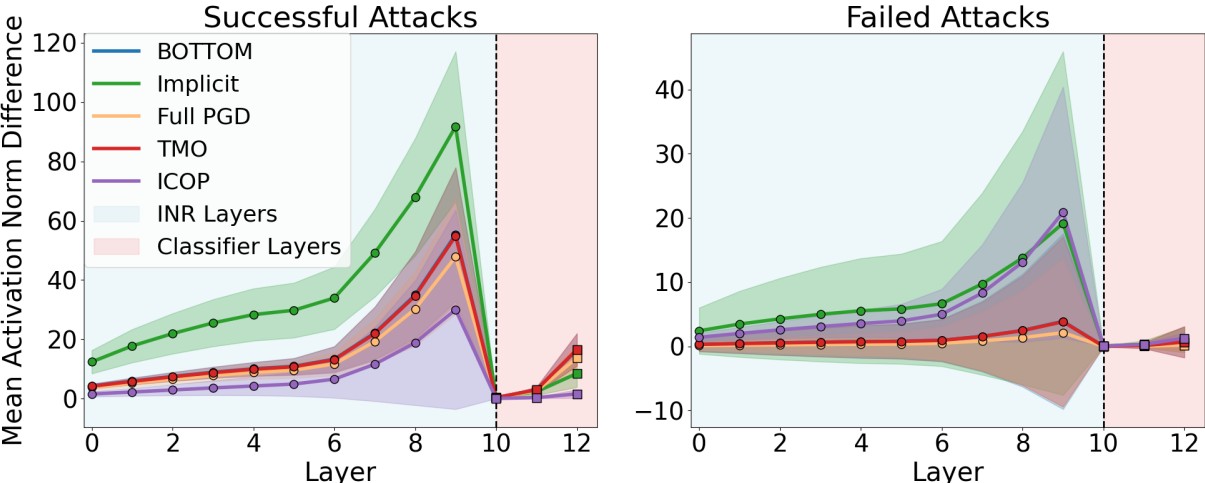

Figure 5: **Adversarial amplification across pipeline layers** - showing modulation optimization's attenuation of adversarial amplification evaluated over the MNIST dataset.

In figure 1 we demonstrate the effect of adversarial perturbations applied in signal-domain over the latent representations attained from perturbed data, and compare between parameter-space classifiers and traditional signal-space classifiers. To this end, as a signal-domain baseline, we first train a convolutional autoencoder over the MNIST dataset. For a fair comparison, this autoencoder is trained disjointly from the downstream classification task. We then train a classifier over the learned latent space, and finally perform 100 iterations of signal-domain adversarial attack over this classifier. The bottom half of figure 1 illustrates t-SNE projection of the trained autoencoder's latent representations before (left) and after (right) the adversarial attack. The top half of the plot corresponds to parameter-space classifiers, showing t-SNE projections of INR modulations optimized for clean (right) data and for data attacked with our full-PGD attack (section 3.4.1). Prior to the attacks, the latent space of both signal and parameter-space domains sis well-structured, with clear separation between the different classes. However, after the attack, the model operating strictly in signal-domain (i.e. the autoencoder baseline) experiences severe disruption of this structure – classes of adversarial samples within the classifier's latent input space are no longer distinct and are instead intermixed. This indicates a significant deterioration in the class separability, highlighting the impact of the adversarial perturbations on the latent space. Adversarial attacks optimized for the parameter-space classifier, on the

other hand, seem inconsequential to the input-space structure of the parameter-space classifier, including class separability. These conclusions are well-correlated with those from figure 5.

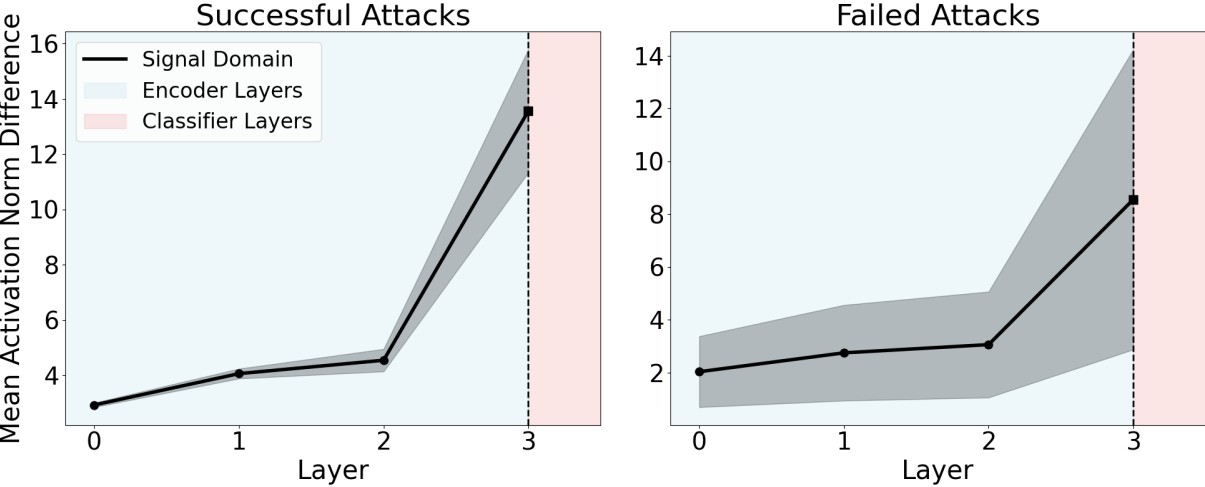

Figure 6: **Signal-Domain Amplification** - through encoder and downstream classifier layers

## 4.4 Theoretical Exploration of the Robustness Mechanism

In this section we wish to justify our hypothesis that the observed relative resistance of parameter-space classifiers to white-box attacks stems from a functional *scrubbing* effect inherent to the INR optimization loop. To achieve this, we provide empirical evidence to substantiate this claim and characterize the security profile of these models against adaptive attacks.

### 4.4.1 Qualitative Analysis

To understand how the INR pipeline processes adversarial perturbations, we analyze the layer-wise amplification of perturbations using the Mean Activation Norm Difference (MAND) metric.

As shown in Figure 5, the parameter-space pipeline exhibits a unique behavior: while perturbations are amplified during the modulation optimization steps (Layers 0–9), this amplification collapses to near-zero at the input of the parameter-space classifier (Layer 10). This strongly-aligns our hypothesis that the INR optimization acts as a low-pass filter (i.e. *"scrubber"*). Due to the spectral bias of the architecture and the limited number of optimization steps, the INR prioritizes fitting the global signal structure while failing to fit the high-frequency adversarial noise. Consequently, the noise is effectively removed from the representation before it reaches the downstream classifier.

In contrast, as discussed in 4.3 signal-domain classifiers exhibit continuous amplification of the perturbation throughout the encoder and classifier layers, explaining their susceptibility to standard first-order attacks.

### 4.4.2 BPDA Analysis

The *"scrubbing"* phenomenon described above implies that the gradient of the loss with respect to the adversarial perturbation, $\nabla_\delta \mathcal{L}$, vanishes as it is backpropagated through the optimization loop. This creates a state of *gradient obfuscation*, where standard gradient-based attacks (like PGD) fail because they rely on informative gradients.

To determine if the model is fully inherently robust or reliant on gradient-masking, we evaluate it against the Backward Pass Differentiable Approximation (BPDA) attack (Athalye et al., 2018). BPDA is an adaptive attack designed to bypass shattered or vanishing gradients. It approximates the gradients of non-differentiable or ill-conditioned layers (in our case, the optimization mapping $R(\cdot)$) by replacing them with a differentiable

approximation-specifically, the identity function-during the backward pass:

$$\nabla_x R(x) \approx I \tag{7}$$

We evaluate the parameter-space classifier against BPDA with 5 random restarts, on both MNIST and Fashion-MNIST across varying perturbation bounds. The results are presented in Table 2.

Table 2: **Robustness against Adaptive Attack (BPDA).** The significant drop in accuracy indicates that the model relies on gradient obfuscation. For any given $\epsilon$ value, the $L_\infty$ attack bound is $B = \frac{\epsilon}{256}$.

|  | Robust Accuracy | | | |
|---|---|---|---|---|
|  | $\epsilon = 4$ | $\epsilon = 8$ | $\epsilon = 16$ | $\epsilon = 32$ |
| **MNIST** | 0.095 | 0.092 | 0.087 | 0.076 |
| **Fashion-MNIST** | 0.167 | 0.163 | 0.16 | 0.159 |

The results in Table 2 reveal that when the gradient obfuscation mechanism is bypassed, the robust accuracy drops significantly (e.g., to 9% on MNIST at $\epsilon = 32$). We denote that the robust accuracy of paramter-space classifiers still remains much higher than that commonly-measured for conventional signal-domain classifiers under PGD or BPDA attacksAthalye et al. (2018). The main conclusion to be drawn from these results is that a valid adversarial direction exists and can be found if the gradient masking is circumvented. Namely, the high robustness against white-box attacks reported in table 1 is a result of the optimization loop attenuating gradients. The main challenge posed for white-box attacks over parameter-space classifiers is *gradient obfuscation*, rather than complete elimination for gradient information useful for adversarial purposes. While this does not provide theoretical security, it imposes a practical barrier, forcing adversaries to employ computationally expensive adaptive attacks rather than standard first-order methods.

Combined with conclusions from drawn in section 4.1, we frame weight-space classifiers as inherently adversarially robust to gradient-based white box attacks, whilst measuring limited and inconclusive robustness to gradient-free attacks (due to gradient obfuscation).

## 4.5 Computational Resource Analysis

In addition to empirical robustness, parameter-space classifiers introduce substantial computational overhead for attackers. These difficulties stem from the optimization process inherent within the classification pipeline's inference process.We measure the per-sample runtime of adversarial optimization on an NVIDIA RTX-2080 GPU and compare it to clean inference. For the MNIST classifier with 512-dimensional modulation vectors and 100 PGD iterations, clean inference requires 1.5 seconds per sample, while attack optimization requires 150 seconds per sample, a $100\times$ increase. Additional results related to the increased memory and time resources required for attack execution are reported in the appendix.

The analysis of runtime required for adversarial optimization is especially relevant when considering Auto-Attack – in table 3 we compare the per-sample execution time of Auto-Attack with that of 100 PGD steps of each attack in our suite. AutoAttack is an ensemble of four attacks — APGD-CE (Croce & Hein, 2020b), APGD-DLR (Croce & Hein, 2020b), FAB (Croce & Hein, 2020a) and Square (Andriushchenko et al., 2020). Since the per-sample execution of each attack component is dependent on its respective predecessor, the runtime indicated in the column of each component is the runtime of executing that component in addition to running all preceding components (for a single sample).

Table 3: **Per-sample runtime (in seconds) for Auto-Attack and the parameter-space suite** - the proposed attack suite is significantly faster than Auto-Attack.

| Attack Suite | Runtime (in seconds) | | | | |
|---|---|---|---|---|---|
| **AutoAttack** | APGD-CE | APGD-DLR | FAB | Square | |
|  | 10.15 | 94.19 | 215.8 | 426.68 | |
| **Parameter-Space-Suite** | Full-PGD | TMO | BOTTOM | ICOP | Implicit |
|  | 9.8 | 2.51 | 10.3 | 6.07 | 13.97 |

Results show that running full Auto-Attack (APGD-CE through Square) would require runtime larger by a factor of 40 when compared to our parameter-space attack suite, while providing comparable adversarial

prowess (table 1). In the best-case (only APGD-CE is run), which is notably less common in Auto-Attack runs against robust models, Auto-Attack runtimes are similar to those of attacks from our suite. We refer the reader to appendix C for a deeper discussion of considerations discussed in this section.

## 5  Discussion

In this work we presented the first thorough systematic study of adversarial robustness in parameter-space classifiers. Our in-depth security analysis positions weight-space classifiers as adversarially robust to white-box gradient-based attacks, especially in comparison to conventional classifiers operating in signal domain. We also mark the limitations of this robust behavior, showing it weakens and potentially fails under the presence of gradient-free attacks (BPDA and black-box components of auto-attack). This leads us to the framing of the gradient-obfuscation phenomenon as a major reason for the observed behavior of increased robustness. The observed robustness is further-stressed by an analysis of the heavy computational costs required for optimizing different types of adversarial attacks over these parameter-space classifiers.

Furthermore, we've presented the challenges involved in translating common PGD-based attacks onto models performing classification directly in deep parameter-space, and addressed these challenges by developing a novel set of five different adversarial attacks designed for parameter-space classification. This work highlights robustness to white-box gradient-based adversarial attacks as a defining feature of parameter-space classifiers and provides a foundation for advancing secure and scalable learning systems.

Despite these promising results, several limitations remain. Parameter-space classifiers are not yet widely deployed in computer vision, constraining immediate applicability. Nonetheless, with the rapid growth of research interest in the field, we expect our findings to encourage the adoption of parameter-space methods. Furthermore, our evaluation focused on datasets where current parameter-space methods achieve competitive clean accuracy, which limited exploration on more complex benchmarks. Finally, our work is focused on inherent robustness for white-box gradient-based attacks. Given the measured inherent robustness measured for gradient-based attacks, we believe the development of active robust training approaches can push this robust behavior to extend to gradient-free and black-box attack methods as well. We defer the exploration of black-box attack paradigms and adversarial pre-training techniques to future work.

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

# A   Training Regime, Hyperparameters & Technical Details

## A.1   Functaset Generation

### A.1.1   MNIST & Fashion-MNIST

All experiments described in this section have been conducted on a single NVIDIA RTX-2080 GPU. Following (Dupont et al., 2022), for each of the three datasets considered in this paper, we first train a SIREN (Sitzmann et al., 2020) jointly over the entire training set. For MNIST and Fashion-MNIST we optimize modulation vectors of dimension 512 for 6 epochs of 3 modulation optimization steps per-sample, with a learning-rate of 0.000005 and a batch size of 128. We use 10 SIREN layers with hidden dimension of 256. We then proceed to creating the Functaset , which is the dataset of modulation vectors conditioning the SIREN for per-sample representation. For each sample we optimize for 10 modulation optimization steps, with a learning-rate of 0.01. For clean training and evaluation we also found it useful to perform gradient clipping to an $\mathcal{L}_2$ norm of 1 during internal modulation optimization.

### A.1.2   ModelNet10

For ModelNet10 we similarly follow (Dupont et al., 2022), however we first convert the Mesh-grid data onto a voxelgrid, as previously mentioned (section 3.5). Each voxelgrid is resampled to dimension $15 \times 15 \times 15$. Due to the increased dimensionality, we train 2048-dimensional Functa modulations with a SIREN similar to the one used for the MNIST dataset. After training the shared SIREN weights, each modulation vector is further optimized for 500 modulation optimization steps.

## A.2   Adversarial Attacks

After the Functaset is generated, we proceed to executing our develop suite of adversarial attacks section 3.4. For MNIST and Fashion-MNIST, all attacks are executed over 10 modulation optimization steps per adversarial perturbation, similarly to optimization in the clean case. BOTTOM and TMO optimization segments are chosen at 5 steps per-sample. For implicit differentiation, we've found it most useful to optimize each perturbed image using L-BFGS, rather than gradient methods (which have been used in other attacks). Each L-BFGS optimization is carried for at most 20 iterations. For ModelNet10, ICOP,TMO and BOTTOM attacks are performed for a total of 500 modulation optimization steps per perturbed voxelgrid, with attack segments again chosen at 5 steps. Implicit differentiation attack is also performed with at most 20 iterations of L-BFGS. All attacks are carried-out for 100 PGD iterations, with early-stop of 3 iterations of constant loss for the implicit method. We use the Adam optimizer in all experiments.

## A.3   Baseline Models

### A.3.1   Classifier Baselines

In this section we specify implementation and training details related to the signal-space classifier baseline CNN and ViT architectures used in section 4. In section 4.1 we train 4 signal-domain classifiers – a small CNN, a ResNet18, ViT-Large/16 and Vit-Base/16. For the ViTs we leverage pre-trained models from (Wightman, 2019), and finetune for our classification task using a linear probe prior to a applying Softmax. For the Resnet we train from scratch, and project encoded data with a linear probe followed by a Softmax. For the small-CNN we use two convolutional layers with ReLU activations, and a single MaxPooling operation between them. For all baselines we train over the Fashion-MNIST training set for 5 epochs with learning-rate of 0.0001.

### A.3.2   Autoencoder Baselines

In section 4.3, we present a qualitative analysis of adversarial perturbation amplification across pipeline layers, along with t-SNE projections of modulation spaces before and after adversarial perturbation. To connect these observations to the inherent adversarial robustness of parameter-space classifiers, we include a comparison with a signal-space classifier. In order to propose a signal-space classifier with an appropriate

counterpart to the explored parameter-space model modulation space, we choose to train classifiers operating over latent spaces trained by autoencoders. We model each autoencoder as a 3-layer CNN with ReLU activations and train with a self-supervised reconstruction objective for 10 epochs. After the autoencoder is trained, we freeze it and train a linear classifier with 2 hidden layers and a single ReLU activation on top of it. The frozen encoder layers acts as the INR counterpart in figure 5, and the MLP head is parallelized to the parameter-space classifier. We use a latent dimension of 128 for all encoders.

## B    Additional Results

### B.1    Robustness For 2D Data

#### B.1.1    MNIST

We explore trends exhibited in figure 5 for additional attack bounds. Figure 7 showcases adversarial amplification (similarly to figure 5) across different attack bound for the full-PGD attack. Adversarial obfuscation drawn from figure 5 arises here as well.

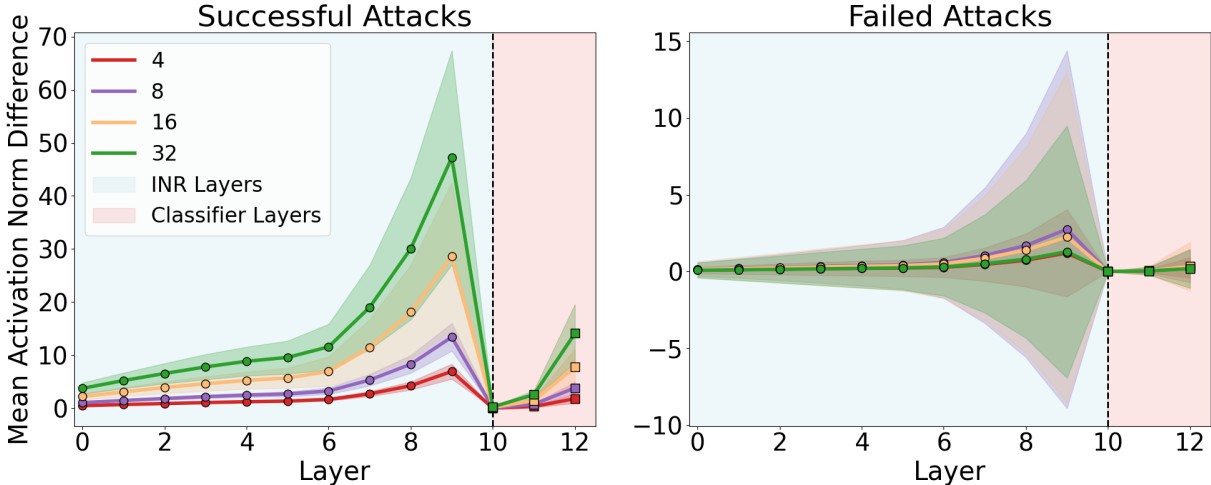

Figure 7: **Signal-Domain Amplification** - through encoder and downstream classifier layers, across different constraints $\frac{\varepsilon}{256}$ for full PGD attack

#### B.1.2    Fashion-MNIST

figure 8 illustrates adversarial amplification across pipeline layers, similarly to figure 5. Trends of correlation between amplification and adversarial success, as well as evident obfuscation of adversarial affect between INR and classifier segments, are consistent with findings from section 4.3.

Similarly to figure 6, we test amplification patterns of a signal-domain classifier (section A.3.2) for the Fashion-MNIST dataset as well. We again observe consistency across both datasets - the signal domain classifiers proves to be much more sensitive to adversarial examples propagated through the encoder-classifier pipeline.

figure 10 exhibits the affect of adversarial attacks in signal-domain over the underlying modulation vectors (for parameter-space classifiers) and latent vectors (for signal-space classifiers) received as classifier input. Similarly to figure 1, for both parameter-space and signal-space model, a well-formed latent space is observed before the incorporation of adversarial perturbation. After attacking with PGD, the corresponding modulation vectors fed into the parameter-space classifier remain almost unchanged, whilst the latent space observed by the signal-domain classifier is heavily deformed.

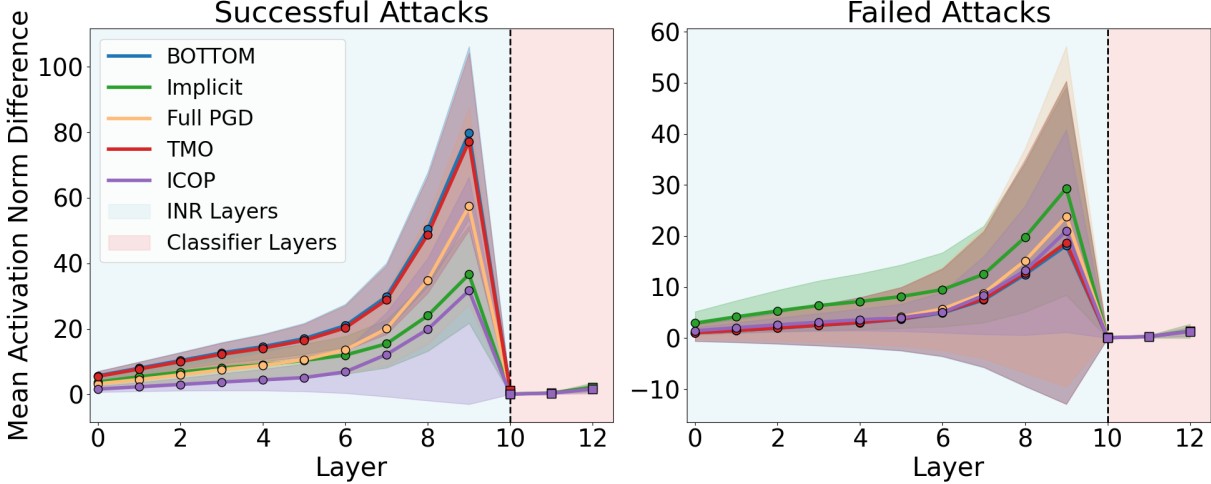

Figure 8: **Adversarial amplification across pipeline layers** - showing modulation optimization's attenuation of adversarial amplification.

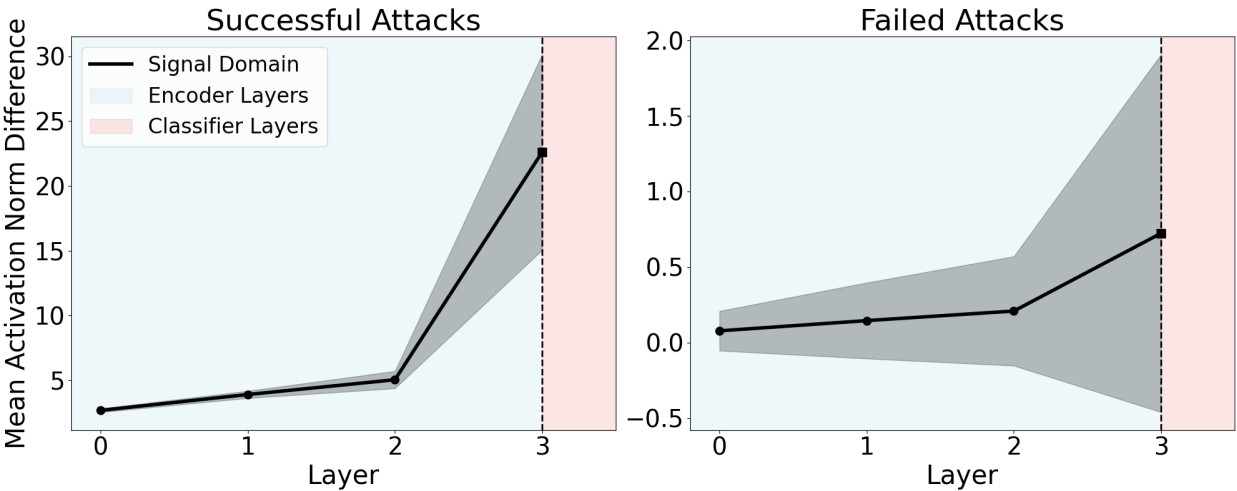

Figure 9: **Signal-Domain Amplification** - through encoder and downstream classifier layers

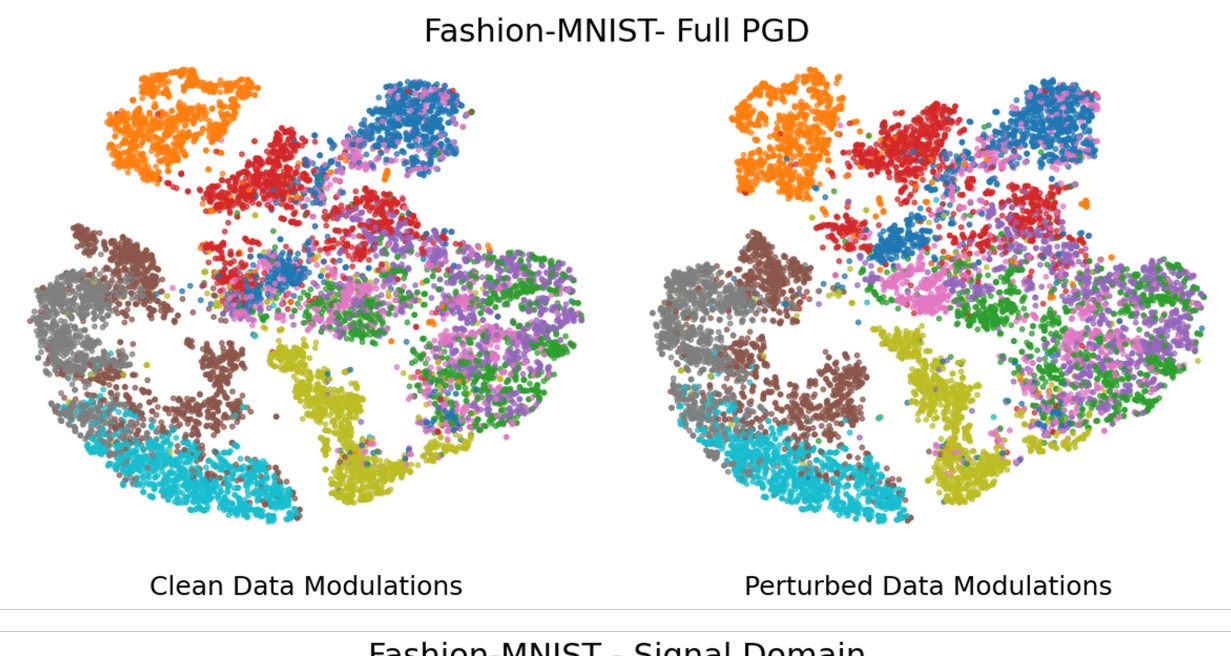

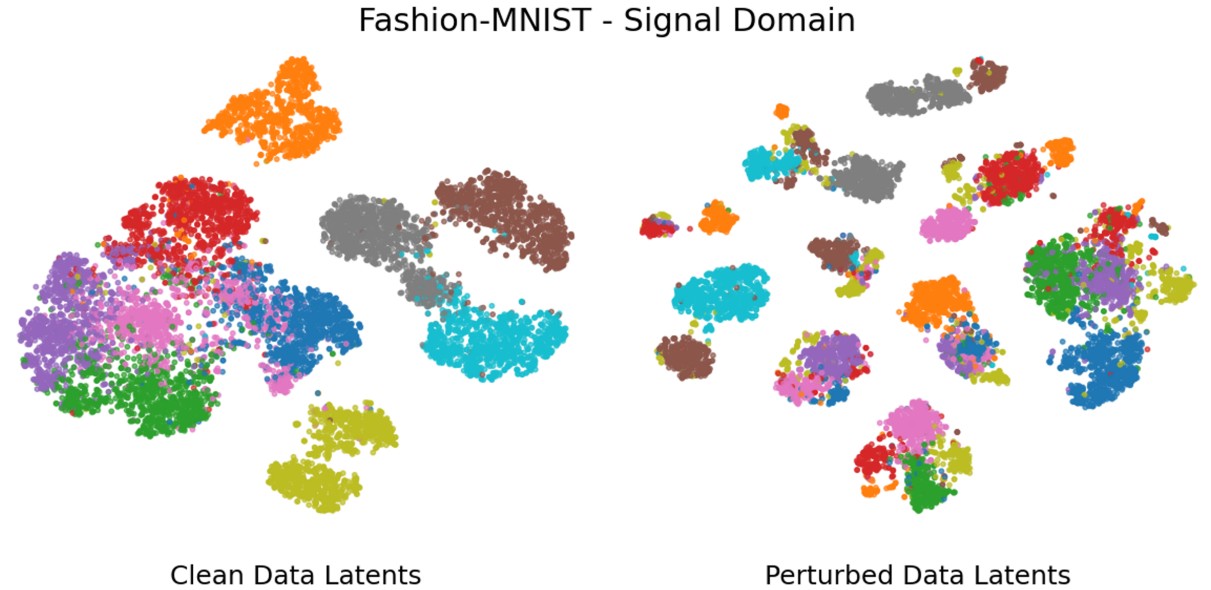

Figure 10: **t-SNE projection of modulation vectors fitted for clean and adversarially-perturbed data** - for parameter-space and signal-space classifiers.

## B.2 Robustness For 3D Data

In this section we report additional results for the ModelNet10 dataset.

In table 4 we report numerical results for the data from figure 4 The first two columns represent the same attack constraints in absolute bit numbers ($\varepsilon$) and bit numbers relative to the entire volume (matching horizontal axis in figure 4).

Table 4: **ModelNet10 Clean/Robust Classification Accuracy** - for any given $\varepsilon$ value, attack bound from equation 3 is $B = \varepsilon$ in $\mathcal{L}_0$ (i.e. number of flipped signal bits).

| $\varepsilon$ | Flip Ratio | Clean | TMO | BOTTOM | ICOP | Implicit |
|---|---|---|---|---|---|---|
| 10 | 0.003 | 0.734 | 0.669 | 0.663 | 0.66 | 0.722 |
| 20 | 0.006 | 0.734 | 0.6475 | 0.634 | 0.6475 | 0.708 |
| 40 | 0.012 | 0.734 | 0.647 | 0.623 | 0.641 | 0.696 |
| 80 | 0.024 | 0.734 | 0.643 | 0.616 | 0.581 | 0.67 |
| 100 | 0.03 | 0.734 | 0.6412 | 0.609 | 0.557 | 0.657 |
| 200 | 0.06 | 0.734 | 0.594 | 0.576 | 0.41 | 0.613 |

## C  Computational Resource Analysis

In sections 4.1,4.2 we provide evidence for the adversarial robustness of parameter-space classifiers based on classification accuracies under the presence of adversarial perturbations. In this section, we expand the discussion from section 4.5 to support these claims from a different, practical perspective.

We show that orthogonally to the efficacy of adversarial attacks over parameter-space classifiers, attacking these classifiers also poses significant added computational difficulties for the potential attacker. In section 4.5 we report the computation times required for a single modulation optimization step during a single PGD iteration on an NVIDIA RTX-2080 GPU (approximately 0.015 seconds for the the 3-layer classifier used for the MNIST experiments, and 0.005 seconds for the single-layer classifier used for the Fashion-MNIST experiments). For the ModelNet10 dataset with 2048-dimensional modulation vectors, the measured runtime is 0.02 seconds. Even for a small number of PGD steps, increasing the number of modulation optimization steps – an aspect controlled by the model, not the attacker – increases attack optimization times within several orders of magnitude. Importantly, this increase does not significantly impact the computation time required for "clean" (unperturbed) inference. For instance, with 100 modulation optimization steps set by the model to solve equation 1, "clean" inference would require 1.5 seconds per sample, while perturbation optimization of 100 PGD iterations would require the attacker to invest 150 seconds per-sample. In table 3 we report the per-sample execution time of Auto-Attack with that of 100 PGD steps of each attack in our suite. showing marked increased costs for running this attack. especially when compared with our developed suite.

On top of computation time increase, attacking parameter-space classifiers via explicit differentiation also poses a challenge for the attacker in terms of memory consumption. Tested on the same hardware, we find that GPU memory consumption for the full PGD attack over the ModelNet10 classifier grows linearly with number of modulation optimization steps, with a slope of $0.1705[\frac{GB}{Mod.Steps}]$. The corresponding coefficient for the smaller MNIST data is $0.049[\frac{GB}{Mod.Steps}]$. Namely, in order to perform a full PGD (equation 3.4.1) attack over the ModelNet10 data, every additional modulation optimization step included in "clean" model training induces roughly 170 additional megabytes for the attacker's backpropagation. Backpropagating through the 500 modulation optimization steps used in creation of the ModelNet10 INR dataset in our experiments would require 85 Gigabytes of GPU memory, greatly increasing optimization costs. The increase in modulation optimization steps holds no more than marginal prices for the "innocent" (non-attacker) user in INR dataset optimization, as this process does not require 2nd-order differentiation. Implicit differentiation (equation 3.4.5) offers the attacker constant-memory attacks (in our experiments - 526 MB for MNIST, 986 MB for ModelNet10), however at the cost of higher sensitivity to optimality condition satisfaction (as shown in section 4.1).

