# OpenReview forum: "Adversarial Attacks in Weight-Space Classifiers"
_TMLR — Accepted by TMLR_

### Review · Reviewer_o9Y9 · 2025-12-01

**Summary Of Contributions:**

The paper proposes adversarial attacks for weight-space classifiers. Due to the optimization step in computing implicit neural representations, standard adversarial attacks can be difficult to be applied. The paper proposes several tailored attacks to this end and evaluates them empirically. When comparing them to adversarial attacks of classic classifiers, the authors find implicit neural representation models to be more robust. However, the validity of this conclusion is not fully established.

**Strengths:**
1. The paper proposes several adversarial attacks in weight-space and introduces them in a logical and well-structured fashion.
2. The experiments appear comprehensive and well-thought out.

**Weaknesses:**
1. The Related Works section is surprisingly short: one would expect some mention of standard adversarial attacks and INR methods.
2. The paper is missing a clear threat model, something that would be expected from a security-adjacent work. For example, the authors chose to attack in signal-domain, before computing the modulation vectors, rather than the modulation vectors themselves (which would allow for much more trivial attacks). But that choice is not explicitly discussed or justified.
3. It is not clear where the optimality conditions in Section 3.3.5 come from, especially the second one. Should it hold for all $dx, d\theta$? If so then the problem isn’t feasible to be solved with gradient-based methods as the loss function would be flat in the vicinity of $\theta^\star$. The value of this formulation of implicit differentiation is further reduced by the fact that the authors end up using L-BFGS instead.
4. A relatively straightforward attack is missing: the attacker can train a surrogate model from signal-domain to weight-space. Then they can apply any standard signal-domain attack to INR.
5. It is very strange that the Full attack is weaker than the BOTTOM and TMO attacks, which are proxies for it. While the authors do speculate that this might be because of vanishing gradients, it doesn’t seem to be actually tested or explored. Actually looking at the gradients would be helpful. Because the same explanation could also cause exploding gradients which the authors do not seem to observe.
6. The paper claims that weight-space classifiers are more robust than signal-domain classifiers. Yet, the results in Table 1 show more nuanced picture: for example, for MNIST, all weight-space attacks have lower accuracy than Auto-attack at $\epsilon=4$ and three of them have lower accuracies for $\epsilon=8,16$. The comparison is further complicated by the clean accuracy of the signal-domain classifiers not being reported.
7. It is difficult to be convinced in the empirical comparison of the robustness of two different sets of models on two disjoint sets of attacks. Unfortunately, 10 years of adversarial robustness research has shown that models are robust only until the right attack and hyper-parameters are found. Unless there are formal guarantees of robustness, a purely empirical study cannot claim robustness.
8. The paper also misses an opportunity to establish why would weight-space classifiers be more robust than signal-domain classifiers. That would help significantly in convincing the reader of the validity of the results. For example, could it be that the explicit down-projection in the INR setup results in the adversarial perturbations to be “scrubbed” before they even get to the model? This could be studied by comparing the original attacked image with the reconstructed version of it: I would expect that the norm of the perturbation of the reconstructed image would be lower than the one of the original image. Thus, the “effective $\epsilon$ could be much lower than the $\epsilon$ that is applied.

Overall, I believe the paper touches upon an important problem and proposes interesting attacks. That being said, it can benefit from rethinking whether its claims that “weight-space classifiers more robust than signal-domain classifiers” are actually backed by empirical results, an attempt to study why that could be the case, as well as an improvement of the writing and the presentation.

**Additional Comments:**

**Nitpicks:**
- Some citations are in double parentheses rather than single parentheses.
- Stray hyphens appear in the beginning of some paragraphs and sections.
- Often the spacing between paragraphs is missing.
- There are some very long and confusing sentences. For example the last sentence of the second paragraph of Section 3.3. Or the second sentence after Figure 4.
- It is not true that adversarial attacks “utilize the highly-nonlinear nature of modern deep classifiers”. A (piece-wise) linear classifier can also be easily attacked. Rather, it is the lack of smoothness that underlies the existence of adversarial examples.
- Not clear if $\theta_w$ is shared among all samples, or across clusters of samples.
- Section 4, “no robust are adversarial training” -> “no robust adversarial training”.
- Section 4.1, missing space before “2-layer CNN”.
- Some strange expressions like “adversarial prowess” or “time resource consumption” (latency?).
- Figures 3 and 4 do not say what dataset the results are on. Figure 5 does not mention neither the dataset nor the model used.
- While the specific setup of the signal-domain attacks is described in Section 4, the setup for the INR attacks is only available in the appendix which makes comparing them difficult.
- After Figure 4: $40x$ -> $40\times$.

**Audience:**

Yes

**Audience Explanation:**

Establishing adversarial attacks and defenses for implicit neural representation models, as well as an understanding of why their robustness properties differ from these of standard classifiers, would be of interest and value to the community.

**Broader Impact Concerns:**

No broader impact concerns beyond the standard disclaimers for adversarial attacks.

**Claims And Evidence:**

No

**Claims Explanation:**

The main claim of the paper is that weight-space classifiers are more robust than signal-domain classifiers. However, the experimental setup makes it difficult to establish whether this is actually the case. See Weaknesses 6 and 7.

**Requested Changes:**

1. **Claim of the paper is that weight-space classifiers are more robust than signal-domain classifiers:** In my view, this claim either has to be substantiated much more extensively and critically, or needs to be dropped altogether.
2. **Explanation of the robustness of weight-space classifiers:** If weight-space classifiers are indeed more robust than classical classifiers, then a strong and experimentally supported explanation of why that would be the case would further strengthen the main claim of the paper.
3.  **Explicit threat model:** Adding an explicit threat model, justifying the design choices that the authors have taken would be helpful for the paper.
4. **Surrogate model attack:** A surrogate model attack would be nice to be added as it is simple and straightforward to implement and might more accessible than the attacks proposed in the paper.
5. **Presentation quality:** The clarity, language and formatting of the paper need to be improved.

---

> ### Author Response · Authors · 2025-12-17
>
> We thank reviewer o9Y9 for a constructive and profound in-depth review of our work. We wish to address several points raised:
>
> Regarding requested change 1 - we thank the reviewer for this important constructive criticism - we agree and have modified the scope and contributions claimed in the paper (we kindly ask the reviewer to refer to first section of our comment  to reviewer uzge for specific details related to this statement).
>
> Regarding requested change 2 - we have added a deeper qualitative and quantitative analysis of the origin of the observed robustness on section 4.4.
>
> Requested change 3 - we added an explicit definition of our threat model (section 3.3).
>
> Requested change 4 - We agree that surrogate model attacks are a potential tool for evaluating robustness, particularly due to their accessibility and ability to bypass gradient masking.
>
> However, we would like to highlight that our proposed TMO attack effectively functions as a dynamic, local surrogate.
>
> A standard surrogate attack typically involves training a separate, static model (e.g., a HyperNetwork or an Encoder) to approximate the mapping $R: \mathcal{X} \to \mathcal{Y}$ across the entire dataset. In contrast, TMO constructs a specific, differentiable approximation of $R(x)$ on-the-fly for each individual sample by unrolling a limited number of optimization steps.
> Furthermore, TMO allows us to bypass the vanishing gradient problem inherent to the full optimization loop (similar to a proxy model) while maintaining high fidelity to the local curvature of the specific input. Additionaly, we argue that TMO is arguably more accessible than a learned surrogate, as it operates directly on the existing model without requiring the training of any external networks or the collection of additional datasets.
>
>
> In essence, TMO potentially provides the benefits of a surrogate (informative gradients, bypassing obfuscation) without the generalization errors associated with training a static proxy model. We acknowledge, however, that investigating transfer attacks from completely different architectures (e.g., attacking an INR classifier using a ResNet surrogate) remains an interesting direction for future black-box robustness studies.
>
> Requested change 5- The paper has undergone additional proofreading by both human readers and designated software. If any points remain unclear or poorly-presented - we would greatly appreciate he courtesy of bringing them to our attention and we will correct them.
>
> Nitpicks - we truly appreciate the thorough list of nipticks - it is very helpful in improving our paper.
> We believe all points have been addressed in the uploaded revised version. We will gladly correct any cases where our corrections are missing or incomplete.
>
> Regarding $\theta_w$ - following the Functa paper which we rely on, this set of parameters is \textbf{shared across all samples in the dataset}. We highlighted this in section 3.1 of the revised version.
>
> Other weaknesses - regarding weakness 3 - we kindly request to refer the reviewer to section 2 of our reply to reviewer uzge.

---

> > ### Comment · Reviewer_o9Y9 · 2025-12-29
> >
> > Thank you for the response and the pointers to the updated manuscript.
> >
> > I do appreciate the addition of the threat model section.
> >
> > However, I do share the concerns of Reviewer uzge that the message of the paper is much more nuanced in the light of the new experiments. I am also not fully convinced that TMO is equivalent to utilizing a surrogate model, precisely because it models only a truncated variant of the model. Weakness 3 also does not seem to be addressed.
> >
> > Regarding the writing, I noticed that the Related Works section is still rather limited, and typos and style issues remain, e.g.:
> > - In the abstract: "to their standard classifiers" -> "to standard classifiers".
> > - Section 3.3: "response of a an" -> "response of an".
> > - Hyphens in the beginning of paragraphs in Section 3.3.

---

> ### Author Response · Authors · 2025-12-30
>
> We thank reviewer o9Y9 for the ongoing discussion and effort towards improving our paper.
>
> We agree with both reviewer o9Y9 and reviewer uzge that our contributions have become more-nuanced. We plan to alter our stated scope of contributions to that written in our second reply to reviewer uzge, in case the reviewers do agree this framing conveys our contributions correctly and responsibly.
>
> We have also corrected several more typing errors raised by the reviewer. The hyphens that still appear on 3.3 are used to emphasize the explanation of the topic header that comes before these hyphens. We prefer to leave them, but if the reviewers jointly believe it looks bad, we will remove them.
>
> We have also added a wider discussion to our related work section.
>
> We apologize weakness 3 has not been answered thoroughly - we will attempt to provide a more detailed answer:
>
> As derived via the Implicit Function Theorem, the gradient calculation relies on a first-order Taylor expansion of the optimality condition around the solution $\theta*$.  This expansion simplifies to the linear system in Equation 4 if and only if the zero-order term vanishes—i.e., $\nabla_{\theta^*} \mathcal{Z} = 0$. If the inner loop is truncated (non-zero residuals), this zero-order term remains, introducing a systematic error term that the standard ID formulation ignores. This mathematically explains our empirical finding that TMO (which tracks the full trajectory) outperforms ID in limited-compute regimes.
>
> To address scope-of-validity: To answer if this should hold for all $d\theta$. The answer is no. This condition holds only at the optimal parameters $\theta^*(x)$ found by the inner optimization loop. It does not imply the loss function is "flat" everywhere, only that we have reached a local minimum. We attribute the relative failure of the implicit differentiation method in our paper (compared to other attack methods we develop) to the difficulty in approximating the original signal (the image) sufficiently well with the INR. This aligns with the general challenge in approximating the original signal well,  known from previous literature in weight-space models
>
> To address the role of L-BFGS: The use of L-BFGS is crucial here. Standard first-order methods (like SGD) often oscillate, leaving large gradient residuals. We employ L-BFGS specifically to drive the gradient norm $\|\nabla_{\theta^*} \mathcal{Z}\|$ closer to zero efficiently, thereby minimizing the error in the stationarity assumption.

---

> > ### Comment · Reviewer_o9Y9 · 2026-01-04
> >
> > I still don’t see how the equation before Eq. 4 represents the conditions for the minimum of $\mathcal L$. Surely the second one should be restricting the Hessian to be positive definite rather than that the gradient should be zero in the vicinity of the local minimum. That being said, considering that you don’t use implicit differentiation, it probably doesn’t matter. But still, you might want to double-check this.

---

> ### Author Response · Authors · 2026-01-05
>
> We appreciate the reviewer's insistence on correct formulation.
> With the help of the reviewer we've identified an ambiguity in our notation in section 3.4.5, and heavily revised this section to present our development and optimality consitions in a way we believe is much clearer.
>
> To clarify to issue with our notation and fixes in recently uploaded revision:
> Equation 4 in question did not represent a sufficient condition for a minimum (which would require a positive definite Hessian). We intended to express the persistence of the first-order stationarity condition ($\nabla_\theta \mathcal{L} = 0$) under perturbation, which is the starting point for deriving the implicit gradient.
> To address this error, we have completely rewritten Section 3.4.5 to remove the ambiguous notation. The text now explicitly derives the update rule via a first-order Taylor expansion of the stationarity condition.
>
> 2. Relevance of Implicit Differentiation - The reviewer noted that "considering that you don’t use implicit differentiation, it probably doesn’t matter.".
> We respectfully disagree, and wish to point out that we do use Implicit Differentiation as a key baseline in our experiments It matters because we use ID as a baseline, and therefore the theoretical validity conditions are critical. As discussed in the revised text, the fact that the ID baseline (which assumes strict stationarity) underperforms our TMO baseline provides the empirical evidence that the stationarity assumption is often violated in finite-compute adversarial settings.
>
> Importantly, we clarify that the error was only in the ambiguous notation presented in section 3.4.5. This correction should now affect the validity of results in any of our ID experiments.

---

> > ### Comment · Reviewer_o9Y9 · 2026-01-06
> >
> > Thank you for the expanded explanation and the corresponding update of the manuscript! I also agree with you that this section is a valuable baseline for the paper.

---

### Review · Reviewer_Zg74 · 2025-12-04

**Summary Of Contributions:**

This paper presents a study to investigate the robustness of parameter-space classifiers against adversarial attacks. The authors first optimize the PGD attack to improve computational efficiency, enabling it to adapt to parameter-space classification setting. Based on this adaptation, they then evaluate the adversarial robustness of parameter-space classifiers. The proposed approach is intuitive and straightforward, and the corresponding experimental results are presented concisely and clearly.

However, there exist numerous deficiencies in experimental evaluation, which limit the strength and generality of conclusions. In addition, the paper shows notable shortcomings in writing rigor and overall readability. Given these issues, the current version of the paper does not meet the publication standards required for TMLR.

**Additional Comments:**

N/A

**Audience:**

No

**Audience Explanation:**

The experiments in this paper fail to demonstrate that the parameter-space classifiers possess robustness against adversarial attacks. Furthermore, the paper does not adequately highlight the specific properties of parameter-space classifiers under adversarial attacks. As a result, the findings are unlikely to provide novel insight or practical value to the TMLR's audience.

**Claims And Evidence:**

No

**Claims Explanation:**

The core claim of this paper is that parameter-space classifiers are robust to adversarial attacks. However, the experiments and analyses of this paper do not provide sufficient evidence to support this claim. The reasons are as follows:

1. The paper only evaluates parameter-space classifiers against the PGD attack (and its computationally efficient variants), which is inadequate for demonstrating adversarial robustness.

2. The experimental evaluation relies primarily on the robust accuracy metric. Although the mean activation norm difference (MAND) is reported, the paper does not compare the MAND of parameter-space classifiers with that of common classifiers, which fails to highlight the robustness advantage of parameter-space classifiers.

**Requested Changes:**

Critical for acceptance:

1. It would be better if the authors could evaluate parameter-space classifiers under more adversarial attack methods, including both classical approaches and recently proposed algorithms. This would help demonstrate the generalizability of the adversarial robustness of parameter-space classifiers.

2. It would be beneficial to provide an analysis to clarify the relationship between MAND metric and robustness accuracy, and compare the differences between common classifier and parameter-space classifier with respect to this metric. This facilitates the analysis of the underlying reasons why the parameter-space classifier maintains high robust accuracy.


Strengthen the work:

1. Several notations are not clearly defined prior to their use, which weakens the readability of the article.

2. The use and definition of some notations are inconsistent, for example, the use of the parameter-space classification model in Eq. (2) and Eq. (4).

---

> ### Author Response · Authors · 2025-12-17
>
> We thank reviewer Zg74 for the thoughtful evaluation of our paper.
> Regarding requested change 1 - We have re-framed our contribution as "relative robustness for standard white-box PGD attacks", since (as correctly mentioned) this is the type of our attacks our evaluation focuses on (even though we also measure relative robustness to Autoattack - which extends beyond our stated contribution and is not strictly white-box or PGD-based). We also added analysis for the theoretical origin of this relative robustness, pinpointed to originate from gradient obfuscation. We hope these two refinements of our work render our featured set of experiments sufficiently-convincing for relative robustness to PGD attacks.
>
> Regarding requested change 2 - We kindly wish to bring to the attention of the reviewer that we have compared to a signal-domain classifier in appendix B in the original submission. In the updated version we have moved this figure and comparative discussion to the main paper (figure 6). We believe the differences between figures 5 and 6 (surveyed in section 4.3) make our MAND analysis more-complete.
>
>
> Regarding unclear notations - we have used human and automatic resources to proofread the paper again and correct several presentation errors. We are highly-motivated to fix any unclarities regarding notations and presentations if there are any remaining. In such case we would greatly appreciate specific pointers.
>
> Stated corrections have been applied via a recently-uploaded revision.

---

### Review · Reviewer_uzge · 2025-12-05

**Summary Of Contributions:**

### Summary

This paper studies adversarial robustness for “parameter-space” (weight-space) classifiers that operate on implicit neural representations (INRs) rather than raw signals. It introduces a suite of white-box attacks that explicitly differentiate through the INR-fitting process (full PGD, truncation-based variants TMO/BOTTOM, an INR-domain attack with orthogonal projection ICOP, and an implicit-differentiation attack), plus a Binary Voxel Attack (BVA) for 3D voxel data. Experiments on MNIST, Fashion-MNIST, and ModelNet10 indicate that parameter-space classifiers exhibit greater empirical robustness than standard signal-space classifiers under comparable perturbation budgets, with additional practical attack costs attributed to the need to differentiate through the inner INR optimization.

Having noted that, the current analysis / evaluation falls short (see "Things to improve" below). I view the paper as a promising first step that needs strengthening the attack evaluation and tightening the implicit-differentiation methodology/assumptions, doing a scholarly job at relating this work with existing nearby approaches (as opposed to over-emphasizing novelty by magnifying into a very niche method), and explaining why the conclusions would generalize to more realistic datasets/settings.

### Things I liked

1. The paper frames and studies a previously under-explored threat model: adversarial attacks against classifiers operating in INR parameter spaces. The authors say un-explored, but can consider a more scholarly review of previous works that tackle similar objectives, such as works that hinder adversarial attacks via INR [RCGH-2022], and highlight what simplifying assumptions they make with their simple autoencoder baselines (as inferred from A.3.2) compared adversarial attacks on the latent space of other neural models [KWT-2022, WCCNRG-2023, LC-2024].
2. If the reported robustness persists under stronger, obfuscation-aware evaluations, this would identify a meaningful defense-by-architecture effect for INR-based pipelines and the proposed attack suite and BVA can become standard tools for this area. Conversely, if robustness is largely due to masked gradients, the paper can still makes a useful contribution by clarifying the evaluation challenges and highlighting where new attack methods (e.g., better implicit differentiation, amortized surrogates, or certified approaches) are needed.

### Things to Improve

1. The central claim of “inherent robustness” is at risk of conflating true robustness with gradient masking/obfuscation arising from the inner optimization loop [QZZZQM-2020]. Hence, I would suggest adding stronger diagnostic attacks, such as BPDA [ACW-2018], EOT [AEIK-2018], BPDA+EOT [TCBM-2020], multi-restart black-box attacks like SPSA/NES (and/or mapping how the proposed ones subsume these). Further, some robustness checks standard in adversarial evaluation are missing, such as (i)gradient-free black-box attacks (do we really need all the white-box attack complexity if we can achieve similar attack-fidelity by black-box attacks), (ii) transfer attacks from surrogate models in parameter space (e.g. attacks trained using gradients from one INR might generalize to others -- see point 3 below), and (iii) multiple random restarts.
2. The implicit-differentiation attack rests on local optimality and invertibility assumptions, that are nontrivial for nonconvex INR fitting. Some mathematical verification of these conditions for when the ID gradients are reliable (e.g., KKT residuals, Hessian regularity) beyond (only) empirical results would be nice. Think of what diagnostics can highlight that the inner problem is sufficiently solved (e.g., small KKT residuals)? This can also lead to interesting insights about trade-offs and conditions under which truncation remains faithful for BOTTOM and TMO like approaches.
3. In equation 3, can you highlight the assumptions required for its validity (invertibility of ∂^2 L/∂θ^2 and quality of the inner solve)? Also, curious if the authors though of using a non-differentiable function to make the implicit differentiation imprecise/approximate at best (or measuring how the robustness changes under BPDA or EOT when the inner INR-fitting loop is treated as a nondifferentiable transformation in the backward pass)?
4. Another class of attacks could be, in similar veins to the clean modulation vector in TMO, is to develop a universal modulation vector, similar to universal adversarial perturbation [MFFF-2017], for the function R and use it in a similar way.
5. The evaluation is limited to relatively simple datasets (MNIST, Fashion-MNIST) and low-resolution voxel grids (15^3); curious why the authors believe these conclusions would generalize well?


### Other improvements

1. The start of the initial paragraphs in each subsection (e.g. 3.3.2, 3.3.3, etc.) is with a `-`. Given this is not standard grammatical formatting, removing them will improve readability. Also, the following paragraphs have no spacing between paragraphs, making it look ugly.
2. Please improve the use of notions for the INR function and how they are used in technical writing. $F$ defined in sec 3.2 and $R$ in sec 3.3 (and 3.3.1, 3.3.2, etc.) doesn't cleanly define how the two relate to one-another. This creates confusion later in section 3.3.4, where the authors casually mention imposing $F(\delta) \leq B$ to equation (2) that as no mention of $F$. Additionally, also improve the writing by revising the second-paragraph in section 3.2 to be more amiable to folks unaware of the verbiage known to people working on this particular topic.

[QZZZQM-2020] Qiu, H., Zeng, Y., Zheng, Q., Zhang, T., Qiu, M., & Memmi, G. (2020). Mitigating advanced adversarial attacks with more advanced gradient obfuscation techniques. arXiv preprint arXiv:2005.13712.

[TCBM-2020] Tramer, F., Carlini, N., Brendel, W. and Madry, A., 2020. On adaptive attacks to adversarial example defenses. Advances in neural information processing systems, 33, pp.1633-1645.

[AEIK-2018] Athalye, A., Engstrom, L., Ilyas, A. and Kwok, K., 2018, July. Synthesizing robust adversarial examples. In International conference on machine learning (pp. 284-293). PMLR.

[ACW-2018] Athalye, A., Carlini, N. and Wagner, D., 2018, July. Obfuscated gradients give a false sense of security: Circumventing defenses to adversarial examples. In International conference on machine learning (pp. 274-283). PMLR.

[RCGH-2022] Rusu, A.A., Calian, D.A., Gowal, S. and Hadsell, R., 2022, June. Hindering adversarial attacks with implicit neural representations. In International Conference on Machine Learning (pp. 18910-18934). PMLR.

[KWT-2022] Kuzina, A., Welling, M., & Tomczak, J. (2022). Alleviating adversarial attacks on variational autoencoders with mcmc. Advances in Neural Information Processing Systems, 35, 8811-8823.

[WCCNRG-2023] Wang, S., Chen, S., Chen, T., Nepal, S., Rudolph, C. and Grobler, M., 2023. Generating semantic adversarial examples via feature manipulation in latent space. IEEE Transactions on Neural Networks and Learning Systems.

[LC-2024] Lu, M. and Chen, B., 2024. On the adversarial robustness of generative autoencoders in the latent space. Neural Computing and Applications, 36(14), pp.8109-8123.

[MFFF-2017] Moosavi-Dezfooli, S. M., Fawzi, A., Fawzi, O., & Frossard, P., 2017. Universal adversarial perturbations. In Proceedings of the IEEE conference on computer vision and pattern recognition. CVP).

**Audience:**

Yes

**Audience Explanation:**

While highlighted in my review above, quoting one line from above to answer this:

If the reported robustness persists under stronger, obfuscation-aware evaluations, this would identify a meaningful defense-by-architecture effect for INR-based pipelines and the proposed attack suite and BVA can become standard tools for this area. Conversely, if robustness is largely due to masked gradients, the paper can still makes a useful contribution by clarifying the evaluation challenges and highlighting where new attack methods (e.g., better implicit differentiation, amortized surrogates, or certified approaches) are needed.

**Broader Impact Concerns:**

Without proper evaluation or deeper understanding, the paper can propagate a false sense of security when using INR-based classification methods.

**Claims And Evidence:**

No

**Claims Explanation:**

The current analysis / evaluation falls short (see "Things to improve" below). I view the paper as a promising first step that needs strengthening the attack evaluation and tightening the implicit-differentiation methodology/assumptions, doing a scholarly job at relating this work with existing nearby approaches (as opposed to over-emphasizing novelty by magnifying into a very niche method), and explaining why the conclusions would generalize to more realistic datasets/settings.

See detailed comments above.

**Requested Changes:**

See the section on "Thinks to improve" and "Other Improvements" in my review.

---

> ### Author Response · Authors · 2025-12-17
>
> We thank reviewer uzge for the detailed and constructive review.
> We have revised the manuscript significantly to address the concerns regarding the validity of the robustness claims and the lack of a mechanistic explanation. Specifically, we have moved from claiming "inherent robustness" to characterizing the system's "relative robustness" as a result of gradient obfuscation inherent to the optimization loop.
>
> 1. Mechanism of Robustness ("Scrubbing") & Validity of Claims - To answer the concern about the origin of our results being gradient obfuscation rather than inherent robustness - we thank the reviewer for the important recommendation to explore deeper-into this matter. We have adopted to proposal to test BPDA in our framework and found that parameter space classifiers are much-less robust to this attack (10-15% robust accuracy, see table 2 in revised upload). In light of this, we reframe our contribution to be the analysis of  "robustness of parameter-space classifiers to standard white-box attacks, relative to their signal-domain counterparts", rather than full "inherent robustness". Nontheless, we stress that our analysis also shows relative robustness for auto-attack (which is not strictly white-box or PGD based).
>
>  We also supply (section 4.4) a deeper analysis of why the observed robustness originates from gradient obfuscation.
> Given these findings, the main impact of our findings is that unlike signal-domain classifiers, parameter-space classifiers require specialized methods for successful white-box attacks.
> We deeply thank the reviewer for helping make our contribution more-accurate and reliable.
>
>
> 2. Concerns related to ID - We agree that the reliability of Implicit Differentiation (ID) gradients hinges on the quality of the inner solution. specifically, that the KKT residuals are negligible ($\nabla_{\theta^*} \mathcal{L}_{rec} \approx 0$) and the Hessian is well-conditioned.While we did not perform a separate scalar analysis of KKT residuals, we believe our existing empirical comparisons (Table 1) serve as a strong proxy validation of these conditions (or rather, their violation).
>
> Empirical Evidence as Diagnostic: As shown in Table 1, the TMO and BOTTOM attacks (which explicitly unroll the optimization trajectory) consistently outperform the Implicit Differentiation attack. If the inner optimization had reached strict local optimality (small KKT residuals), the ID gradients would theoretically converge to the exact gradients (or match the unrolled gradients). The fact that ID underperforms suggests that in realistic adversarial settings - where the attacker is constrained to a finite computational budget (e.g., 20 L-BFGS iterations)-the strict stationarity required for ID is not fully satisfied.
>
> We observe a trade-off where when the inner problem is non-convex and only "partially" solved (common in adversarial loops), truncation-based methods (TMO) are more faithful because they compute the gradient of the optimization path actually taken, rather than the gradient of a theoretical fixed point that was never reached.We have updated Section 4.4 (Discussion of Results) to explicitly discuss this trade-off, clarifying that the performance gap between TMO and ID highlights the fragility of ID assumptions in non-convex INR fitting.
>
> 3. Validity assumptions - The validity of the gradient approximation in Equation 4 rests on two conditions derived from the Implicit Function Theorem:
> First, the inner optimization must be solved to convergence, such that the gradient of the reconstruction loss is negligible.
> Second, the Hessian of the inner objective must be non-singular at the optimal solution $\theta^*$.
> We have not explored a non-differentiable surrogate for ID - we agree that our ID attacks (and other attack formulation) can be further-explored and adapted for our specific domains. We defer the exploration of this interesting possibility for future work.
>
> 5. Regarding the transfer of findings to more-complex datasets - we believe our findings are transferable because as the data becomes more complex (e.g. more image features or voxels for the INR to represent), the optimization process of the INR becomes more complex. Namely, more gradient steps would be required in $R$ to find the correct INR. We strongly believe this would aggravate the  gradient obfuscation issue encountered by the attacker. Initial evidence lies in our transition from 2D to 3D experiments. There, instead of 20-100 modulation optimization steps, hundreds were needed. This deepens the phenomenon we claim promotes relative robustness via obfuscation.
>
> Other improvements - the suggestion under 1 (removing hyphens) has been applied. We also tried elaborating on the meaning of Functas and SIRENs in 3.1 as-per comment 2. We kindly ask the reviewer to let us know if they feel this issue had not been resolved.
>
> We have uploaded a revised version of the paper, with all stated corrections.

---

> > ### Comment · Reviewer_uzge · 2025-12-24
> > **Rethinking contributions in a more holistic sense**
> >
> > I would like to thank the authors for the additional effort. The newly added section 4.4.2 definitely helped me clarify a few things, which I originally suspected, better. Now, the elephant in the room is where do the contributions stand.
> >
> > In my mind, the paper is proposing a defense mechanism (weight-space classifiers) and four attack methods targeted for this defense mechanism. As part of the review, we now understand that the benefits seen from the defense mechanism was due to different concept (gradient obfuscation) and that there already exists an class of attacks (e.g. BPDA, EOT, etc.) which can break the proposed defense (and questions the need for the newly proposed attacks). Given this
> > 1. Authors will need to describe this holistic narrative better (given the defense is _already_ broken) in the abstract and the introduction instead of selectively highlighting only the positives (e.g. "robustness of parameter-space classifiers to standard white-box attacks, relative to their signal-domain counterparts").
> > 2. Do we view this more as a negative result work OR do you see ways in which this defense can be _fixed_ or _tweaked_ to address BPDA-like attacks?
> >
> > Such publications are definitely important from security standpoint (think cyber-sec/cryptography); in principal, good negative results with proper evaluation give a holistic picture to the community on how to consider evaluations for novel defenses, I would love to hear the perspective of other reviewers and action editors on this.

---

> ### Author Response · Authors · 2025-12-26
>
> We thank the reviewer for the pivotal discussion. We agree that the paper has evolved from a "defense proposal" to a "security characterization of a distinct architecture."
>
> To address the request for a holistic narrative, we believe the paper raises both negative and positive results in terms of robustness in parameter-space classifiers. We propose framing the contribution not as an entirely failed defense, but as a characterization of the security profile of parameter-space classifiers. This profile is defined by three distinct layers of resistance that we believe that are mapped-out in the paper.
>
> 1. White-Box Immunity - First, we establish that these classifiers are naturally immune to the vast majority of "off-the-shelf" gradient-based white-box attacks (like standard PGD). This is a non-trivial property. In standard literature of signal-space classifiers (without robust training), such attacks bring performance to almost 0 on datasets like MNIST and Fashion-MNIST. By contrast, our architecture neutralizes this "white-box privilege" via obfuscating gradients, forcing the attacker to abandon standard tools.
>
> 2. Even when an attacker employs adaptive methods (BPDA) to bypass the gradient obfuscation, the defense does not collapse completely (i.e. robust accuracy is 9%-16% which is much lower, but not as low as the almost-zero robust accuracy seen in signal-space classifiers).
>
> We believe this indicates that the "Scrubbing Effect" of the inner optimization loop is not just a gradient mask - it functionally attenuates the attack signal. While not a certified defense, it creates a soft robustness floor that is significantly higher than the total fragility of standard models.
>
> 3. Cost - we highlight the "Practical Robustness" imposed by the architecture. As detailed in Section 4.5, attacking this pipeline increases latency by two orders of magnitude (~100x). Combined with the need for adaptive strategies, this imposes a severe cost on the adversary, that appears in BPDA as well.
>
> To conclude, we agree with the "negative result" narrative-shift for inherent robustness. However, we view the characterization of this "Gradient Obfuscating, Computationally Expensive, Residually Robust" profile as a significant positive contribution to the security literature. It prevents the community from mistaking obfuscation for certified robustness while accurately mapping the unique costs this architecture imposes on attackers.
>
> We would appreciate reviewer uzge's opinion (and that of other reviewers)  on this modified framing. In case the reviewers generally agree, we will revise the paper to explicitly frame this "holistic narrative," ensuring that the vulnerability to BPDA is presented alongside the residual robustness and computational hardness, offering a balanced view of the architecture's security properties.
>
> We are also open to framing our paper as a negative result entirely, in case reviewers reject our framing proposed here.

---

> > ### Comment · Reviewer_uzge · 2025-12-28
> > **I am okay with the the reframing of the work**
> >
> > I like the framing suggested by the authors.
> >
> > This way, it highlights that the approach is a reasonable defense against certain attacks (and when the threat model only allows for such attacks, it could be useful), but is broken in a more general sense while keeping alive the possibility that there might be some element of weight-space classifiers beyond viewing it as gradient-masking that provides it some (although little) robustness. Another useful aspect is that most other defenses will use auto-attacks to test their defenses, the addition experiment done by the authors will show that reimagining the defense from a slightly different standpoint can highlight gaps in the threat model that only consider popular template attacks.

---

### Author Response · Authors · 2026-01-01

We thank all reviewers for the thorough and constructive evaluation of our work, and for the ongoing helpful discussion.
Many comments raised by the reviewers had been immensely helpful in improving weakness points the paper, especially those related to readability and scope of stated contributions.

We have uploaded a final revision of our work, which we sincerely hope answers all concerns raised by the reviewers. The main modification   from previous revisions raised (discussed in specific reply comments to each reviewer) is that re-framing of our contributions - from inherent robustness of parameter-space classifiers, to a security=analysis paper of such classifiers, that finds two main conclusions - 1. Parameter-space classifiers are robust to white-box gradient-based attacks, especially when compared with conventional classifiers.
2. This robustness mostly stems from gradient-obfuscation, rendering lessened robustness to other (gradient-free) approaches, such as BPDA or Auto-Attack (which is partially black-box).

---

### Decision · Action_Editor_ri1g · 2026-02-04

**Recommendation:** Accept as is

**Audience:**

Yes

**Audience Explanation:**

All reviewers agree that the findings are of interest to the TMLR audience. The paper explores a new class of weight-space defenses and, through its revised analysis, serves as an instructive example of identifying and mitigating gradient masking. Reviewers specifically highlighted the value of this work as a rigorous "negative result," clarifying that the robustness of Implicit Neural Representations stems primarily from optimization-based obfuscation and computational cost rather than certified security. The community will benefit from both the specific exploration of this defense mechanism and the demonstration of rigorous security evaluation practices.

**Claims And Evidence:**

Yes

**Claims Explanation:**

All reviewers agree that this paper has enough evidence to support its claims. Following a productive rebuttal phase, the authors incorporated feedback regarding gradient masking and tested additional attacks against their method. The resulting manuscript offers a more honest assessment of the proposed defense's capabilities and limitations. As mentioned by one reviewer: "by transparently reporting these findings, the paper now highlights good security practices and encourages the community to holistically test defenses before making broad robustness claims."